# Recent centennial drought on the Tibetan Plateau is outstanding within the past 3500 years

Yu Liu [1,2] ✉, Huiming Song [3] ✉, Zhisheng An [1,2], Qiang Li [3], Steven W. Leavitt [4], Ulf Büntgen [5,6,7], Qiufang Cai [1,2], Ruoshi Liu[3], Congxi Fang [8], Changfeng Sun [3], Kerstin Treydte [9,10], Meng Ren[11], Lidong Mo [12], Yi Song[1], Wenju Cai [13], Quan Zhang[1], Weijian Zhou [1,2], Achim Bräuning [14], Jussi Grießinger [14,15], Deliang Chen [16], Hans W. Linderholm [16], Ashish Sinha [17], Hai Cheng [3], Lu Wang[18], Ying Lei [1], Junyan Sun [1], Wei Gong[19], Xuxiang Li [3], Linlin Cui[20], Liang Ning[21,22,23], Lingfeng Wan[24], Thomas W. Crowther [12] & Constantin M. Zohner [12]

Given growing concerns about global climate change, it is critical to understand both historical and current shifts in the hydroclimate, particularly in regions critically entwined with global circulation. The Tibetan Plateau, the Earth's largest and highest plateau, is a nexus for global atmospheric processes, significantly influencing East Asian hydroclimate dynamics through the synergy of the Asian Monsoon and the Westerlies. Yet, understanding historical and recent hydroclimate fluctuations and their wide-ranging ecological and societal consequences remains challenging due to short instrumental observations and partly ambiguous proxy reconstructions. Here, we present a precisely-dated 3476-year precipitation reconstruction derived from tree-ring $\delta^{18}O$ data on the Tibetan Plateau, representing one of the few multi-millennia-long annually-resolved terrestrial $\delta^{18}O$ records to date. Our findings reveal that the 20th century drought extremes are severe within the past three millennia, and likely linked to the weakening of both the Asian Monsoon and Westerlies due to anthropogenic aerosol emissions. Additionally, our analyses identified three distinct stages (110 BC–AD 280, AD 330–770 and AD 950–1300) characterized by shifts toward arid hydroclimate conditions, corresponding to significant social unrest and dynasty collapses, which underscores the potential societal impacts of severe hydroclimatic shifts.

Hydroclimate can significantly impact ecosystems and human well-being through the occurrence of droughts and pluvials[1,2]. The Asian Monsoon (AM), characterized by its pronounced seasonal oscillation between dry and wet seasons, is one of the Earth's most influential atmospheric circulation systems, and particularly vulnerable to future changes in climate[3]. As a crucial water source for 60% of the world's population, any disruptions of the AM could lead to widespread droughts, floods, and other disasters[3], threatening human well-being.

Over recent decades, paleoclimate records, including tree rings, stalagmites, and lake sediments, have provided evidence that spatial-temporal fluctuations in AM precipitation patterns had profound implications on human societies by affecting water resources and

agricultural productivity[4–6]. Yet, establishing a tangible link between fluctuations in the AM and human well-being remains a key challenge, especially during the Late Holocene−a period marked by rapid societal progress. This difficulty arises from the scarcity of long-term, accurately-dated, and annually-resolved paleoclimatic data on AM hydroclimatic changes. Given the critical implications, there is a pressing need for comprehensive hydroclimate reconstructions that accurately capture the natural variability of the AM system.

Monsoonal rainfall in East Asia is closely connected to the mid-latitude Westerlies, which are greatly modulated by mountainous landscapes, such as the Tibetan Plateau and the neighboring mountains[7–10]. However, the mechanisms underlying these large-scale effective connections across various time scales remain poorly understood, primarily because of the lack of reliable hydroclimate records. The Tibetan Plateau, with an average elevation of more than 4000 m a.s.l., plays an important role in regulating the interactions of the Westerlies and the AM[9,10]. Therefore, it is a core region for exploring the spatially as well as temporally varying interplay between these two climate systems.

As a long-living, moisture-sensitive tree species on the Tibetan Plateau, the Qilian juniper (*Juniperus przewalskii* Kom.) offers a unique opportunity for paleoclimate research and establishment of climate reconstructions for the region (Fig. 1). Ancient wood samples from these trees, discovered in old tombs, further allow us to establish tree-ring chronologies spanning the late Holocene[11]. The oxygen isotope ratios in cellulose ($\delta^{18}O$) from tree rings capture annual climate information[12], offer insights over a wide geographic context, and are minimally affected by topography[13]. Since tree-ring $\delta^{18}O$ is significantly dominated by the source water $\delta^{18}O$[14], its analysis enables investigation of details of large-scale atmospheric circulation patterns[15,16], allowing for comparisons with moisture records derived from other paleoclimate data, such as stalagmites, lake sediments, and ice cores[17]. Such multi-parameter approaches are essential to comprehensively understand historic climate shifts and tipping points and their broader implications for societal progress.

Here, we present a well-replicated tree-ring $\delta^{18}O$ chronology with a continuous annual resolution from the Tibetan Plateau covering the last 3.5 millennia (Fig. 2). This long-term record offers the unique opportunity to quantify and understand precipitation fluctuations on annual to centennial time scales. Our study therefore contributes to the understanding of hydroclimate dynamics that have shaped the Tibetan Plateau over recent millennia, with respective implications for human civilization in the region.

## Results and discussion

### Tree-ring $\delta^{18}O$ chronology from the Tibetan Plateau

The Tibetan Plateau tree-ring $\delta^{18}O$ chronology was constructed from Qilian juniper cores and disks collected from living and dead trees, as well as from ancient wood found in tombs across the northeastern Tibetan Plateau. Following successful cross-dating, alpha-cellulose was extracted from individual rings of 17 wood samples and analyzed (Supplementary Table 1). A total of 15,028 individual tree-ring $\delta^{18}O$ values were measured, revealing high inter-series correlations among the 17 individual tree-ring $\delta^{18}O$ series (Supplementary Table 2). For each year between 849 BC to AD 2010, data from at least four trees were available, ensuring a statistically and representative robust dataset. The average segment length of all samples was 885 years, with two samples exceeding 1500 years and 11 over 800 years in length (Fig. 2a, d), which helps to reduce noise from isotopic differences between individual samples and to maintain high- and low-frequency climate signals. The consistency and accuracy of cross-dating among all tree-ring $\delta^{18}O$ series were validated by the values of the expressed population signal (EPS) and the mean inter-series correlation (Rbar)[18] (Fig. 2b).

Due to variations in microclimate or altitude, trees may exhibit varying average tree-ring $\delta^{18}O$ values. To detect potential offsets among trees resulting from different sample origins, we employed an "outliers correction method"[19], which did not identify any outlier trees in our study (see Methods & Supplementary Note). Considering the similarity of tree-ring $\delta^{18}O$ values among various trees during the common period, we established the final tree-ring $\delta^{18}O$ chronology (Fig. 2c) covering a span of 3476 years (1466 BC–AD 2010), by arithmetically averaging all individual $\delta^{18}O$ series[20].

### Long-term precipitation reconstruction on the Tibetan Plateau

Our comprehensive 3476-year precipitation reconstruction on the Tibetan Plateau was achieved through climate-proxy calibration using a regional climate dataset. This dataset was developed by scaling

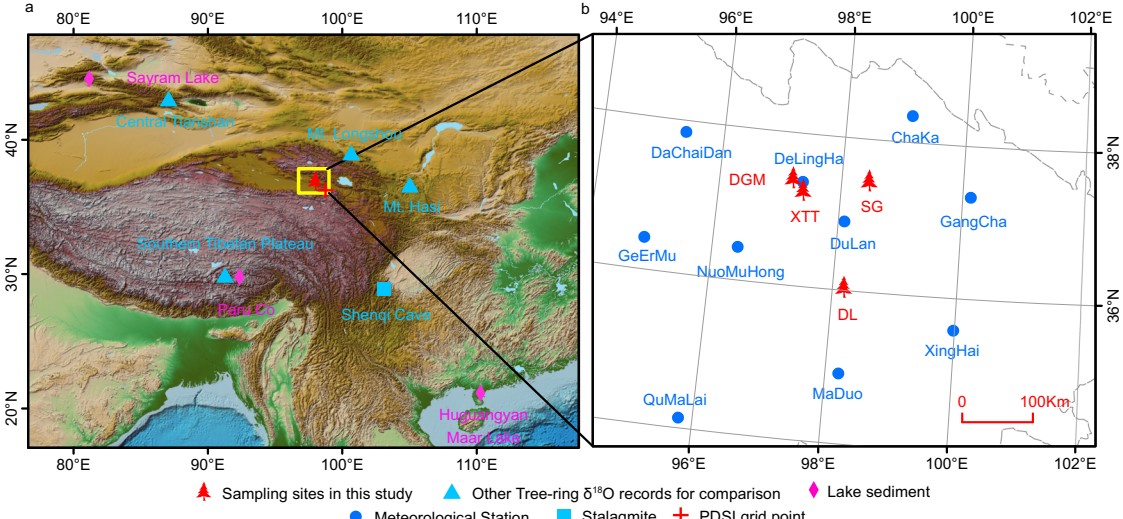

**Fig. 1 | Geographical overview of proxy and meteorological data locations in High Asia used in this study. a** Topographical map highlighting the investigation region, which is the focal area for tree-ring sample collection demarcated by a yellow rectangle. Additional paleoclimate proxies, including lake sediments and speleothem (stalagmite/cave) records, are indicated for context and cross-archive comparison. **b** Detailed map of the study region shown in panel a, pinpointing the positions of the ten meteorological stations and the four primary tree-ring sampling sites (Supplementary Table 1). The base maps are shaded relief images of the ETOPO1 Global Digital Elevation Model.

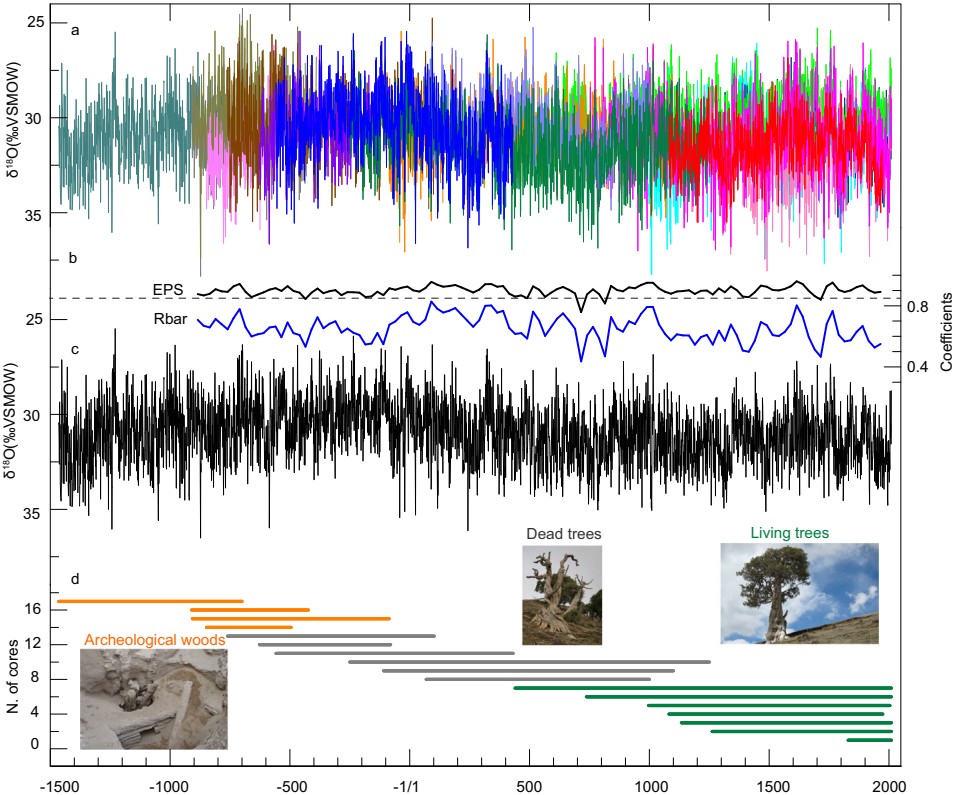

**Fig. 2 | Composite tree-ring δ¹⁸O chronology from the northeastern Tibetan Plateau spanning 3476 years. a** Composite of the 17 raw annually resolved tree-ring δ¹⁸O series depicted with different colors for visual clarity. **b** Dual indicator of the tree-ring data reliability: the Expressed Population Signal (EPS) and the mean inter-series correlation (Rbar). Each metric is calculated over a 50-year moving window, offset by 25 years, to assess the signal strength and coherence among tree-ring sequences. Dashed line represents an EPS value of 0.85. **c** Master tree-ring δ¹⁸O chronology for the Tibetan Plateau, which is the mean value derived from all individual series, providing a continuous 3476-year record. **d** Temporal distribution of each tree's contribution to the final composite δ¹⁸O chronology. Individual lines represent the lifespan of each tree, with orange indicating archaeological wood samples, gray representing dead trees, and green living trees, thereby illustrating the overlap/crossdating of individual tree-ring series across millennia.

observed data from ten meteorological stations (Fig. 1b, Supplementary Fig. 1) located in the study region (Methods). The tree-ring δ¹⁸O data correlated strongly with the sum of regional precipitation from the previous September to August of the current year ($P_{S-A}$), explaining 61% of the variance ($r = -0.78$, AD 1961–2010, $p < 0.001$, Fig. 3a, b, Supplementary Fig. 2 and Table 3). The tree-ring δ¹⁸O is mainly dependent on source water δ¹⁸O and local evaporation fractionation. In our study region characterized by arid climate, the source water is predominantly determined by precipitation. Decreases in precipitation, typically associated with increasing aridity, may increase evapotranspiration and consequently lead to higher ¹⁸O enrichment in leaf and soil water due to elevated atmospheric vapor pressure deficit[14,21]. Therefore, the tree-ring δ¹⁸O record can reflect precipitation variability on the northeastern Tibetan Plateau.

Given the strength of this relationship, we applied a transfer function that employs tree-ring δ¹⁸O as an independent variable (Methods), to reconstruct $P_{S-A}$ for the past 3476 years (Fig. 3d). This reconstruction closely aligns with observed $P_{S-A}$ data (Fig. 3c). A split calibration-verification analysis (Supplementary Table 4) confirmed the reliability of our reconstruction, further supporting the accuracy and quality of the reconstructed precipitation data.

Yang et al. (2021) have developed a 6700-year long tree-ring δ¹⁸O chronology which is situated near to our study site[6]. Therefore, their results were utilized as a reference for validation and verification with our reconstruction. The comparison (Supplementary Fig. 3) revealed low-frequency similarities between the two series, further confirming the overall reliability and representativity of our data set. Specifically, the first six hundred years (1466–850 BC) of our chronology, which are

based on a single tree replication only, also exhibited significant coherences with Yang's series (Supplementary Fig. 3a), making this period valuable as a reference in the absence of other available annually resolved data. However, caution should still be exercised when interpreting our precipitation data for the period during 1466–850 BC due to the limited replication. Additionally, discrepancies were observed between the two tree-ring δ¹⁸O chronologies, which could be attributed to differences in resolution and data composition. Our dataset is annually resolved and based on individual trees, while most of Yang's dataset has a resolution of 3–5 years and was developed using a pooling approach of contemporaneous tree rings from different tree individuals during recent millennia. Furthermore, considerably greater variability is observed during periods where samples were pooled compared to other periods.

Our precipitation reconstruction for the Tibetan Plateau offers critical insights into the changing hydroclimate conditions throughout the late Holocene. On an annual scale, we found a significant correlation between our precipitation series and those derived from local tree-ring-width measurements[11] ($r = 0.32$, $n = 3476$, $p < 0.0001$). However, our reconstruction differs in that it is not influenced by age-related trends. While our tree-ring δ¹⁸O chronology reveals a persistent long-term hydroclimate trend—specifically, a shift from wet to dry conditions—this trend is not reflected in the local tree-ring width chronology[11]. This discrepancy may be attributed to the statistical detrending method used to remove age-related growth trends in tree-ring widths, which could inadvertently also remove the long-term climate trend conserved in tree-ring width series. This highlights the superiority of using tree-ring δ¹⁸O over traditional methods based on tree-ring width and

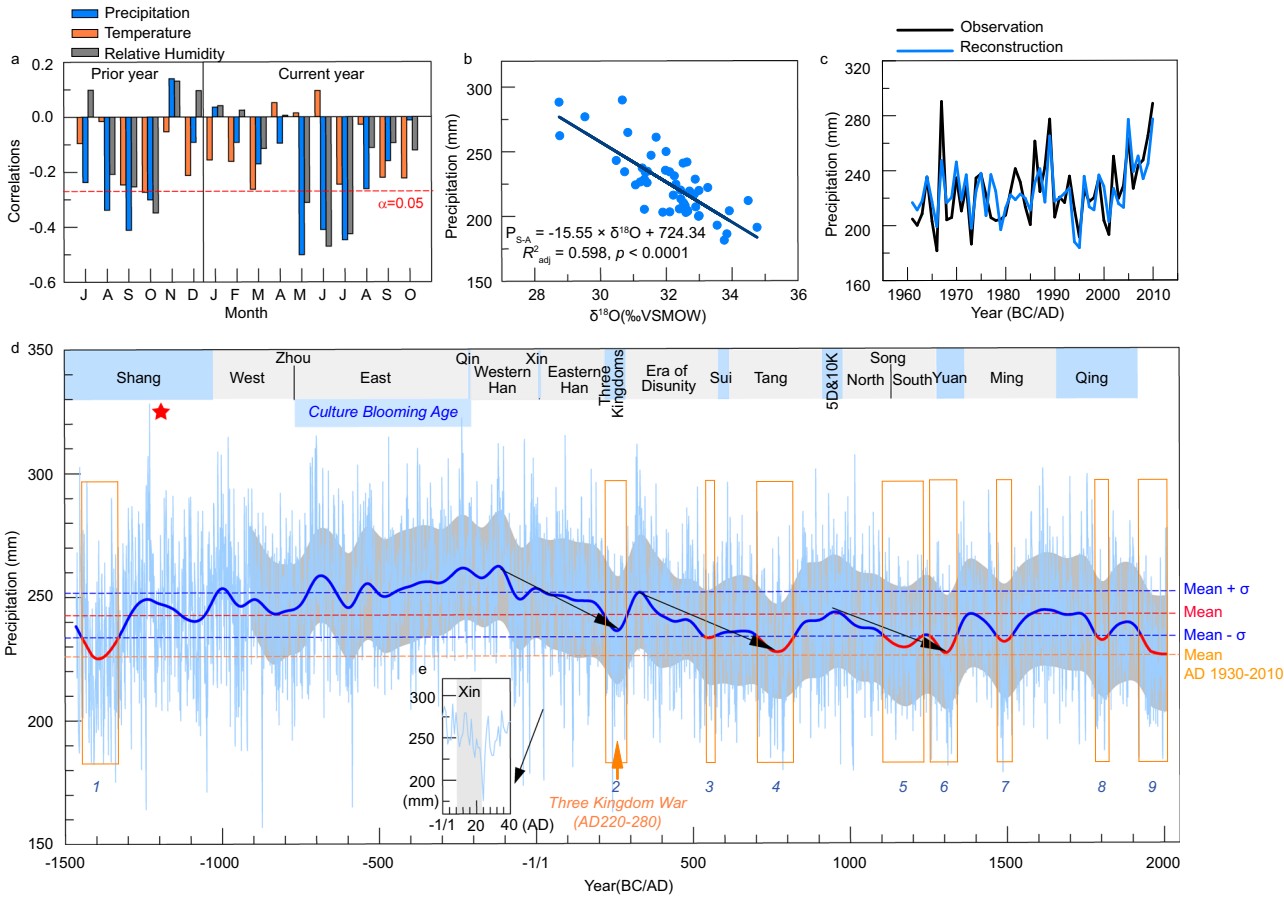

**Fig. 3 | Annually resolved hydroclimatic variability from tree-ring δ¹⁸O on the northeastern Tibetan Plateau over 3476 years. a** Correlation of the tree-ring δ¹⁸O chronology with monthly precipitation, temperature, and relative humidity spanning from the previous July to the current October (AD 1960–2010). Bars indicate the strength and direction of the correlations for each month, with significance denoted by the dashed line at $\alpha = 0.05$. **b** Relationship between the annual precipitation totals from previous September to current August and corresponding tree-ring δ¹⁸O values, underpinning the precipitation reconstruction model. **c** Time series comparison of observed and reconstructed annual precipitation ($P_{S-A}$) demonstrating the accuracy of the tree-ring δ¹⁸O-based reconstruction against instrumental records. **d** Long-term annually resolved 3476-yr precipitation reconstruction for the northeastern Tibetan Plateau (thin blue line), juxtaposed with a 100-yr low-pass filtered representation (thick blue line) to highlight broader

climatic trends. The overall mean of the reconstruction is depicted by the horizontal red dashed line, and blue dashed lines indicate the bounds of one standard deviation from the mean. Notable long-term drying trends are marked by black arrows. Historical Chinese dynasties are chronologically annotated at the top, providing historical context to the climatic data. Bold blue numbers represent nine severe drought periods identified from the 100-yr low-pass filter. The light gray shading in (**d**) represents the ±1σ error range of the record after 100-yr filtering, and the star marks the 13-yr maximum in precipitation across the entire reconstruction. Red segments along the 100-yr filter line denote periods of extreme drought where precipitation fell below the mean-1σ threshold, labeled from 1 to 9, with event 3 being historically significant despite not falling below the mean-1σ threshold. **e** Magnified view of the curve for Xin Dynasty (AD 9–23). Source data are provided as a Source Data file.

density[22] series for long-term climate reconstructions, thereby providing a more robust and reliable climate reconstruction.

To explore hydroclimate variations over the past 3476 years on multi-decadal to centennial scales, we applied a 100-yr low-pass filter to the precipitation reconstruction. This analysis revealed that the average precipitation from the 720 s BC to the AD 70 s was consistently more than one standard deviation above the mean of the entire reconstruction, indicating the longest continuous pluvial period within the past three millennia. Similar prolonged wet periods were also identified in other regions, including West Africa[23], Central Europe[24], and Central Asia[25], suggesting widespread positive hydroclimate anomalies during these times. Another notable wet period occurred during the AD 290 s to 390 s, which was followed by a progressive drying trend, with precipitation levels mostly falling below the long-term average. Our analysis highlights nine severe drought periods, including 1446–1331 BC, AD 220–280, AD 538–568, AD 706–824, AD 1111–1232, AD 1256–1341, AD 1470–1517, AD 1788–1824, and AD 1930–2010 (numbered 1 to 9 at the bottom of Fig. 3d, Methods). Despite a high uncertainty for the drought period during 1446–1331 BC

(Fig. 3d), the drought conditions of the 20th century (1930 to 2010) are severe within entire 3476-year record (Fig. 3d), with eight of the top 100 driest years (8%) occurring within it. By contrast, only one, four, and two of the top 100 driest years were recorded in the 17th, 18th and 19th centuries respectively. Notably, no years within the period of 588–44 BC rank among the top 100 driest years. Furthermore, it is worth mentioning that none of the years since AD 1756 are included among the top 100 wettest years, thereby emphasizing a discernible trend towards aridity in recent centuries.

The extreme droughts of the 20th century are corroborated by other regional paleoclimate reconstructions, including a tree-ring-width series from the western margin of Asian Summer monsoon[26], stalagmite records from the southeastern Tibetan Plateau[27], and lake sediment records from Central Asia[25]. These sources collectively indicate the nature of 20th-century drought period. Particularly in China, the late 1920s saw large-scale severe droughts that corresponded with millions of deaths due to harvest failures and starvation[28]. The consistency of the generally underlying drying trend emphasizes the profound impact of increasing frequency of extreme climatic events on human societies.

Previous simulation efforts have shown that anthropogenic aerosols play a significant role in the observed decline of precipitation in the marginal areas of the Asian Summer Monsoon[26]. To further explore the impact of human activities on hydroclimate variability in this study, we used modeled data from the Community Earth System Model Last Millennium Ensemble (CESM-LME) project[29]. The CESM-LME has been found to realistically simulate the multiple-scale responses of temperature and Asian summer monsoon precipitation over the Tibetan Plateau to external forcings during the last millennium[30–32]. The model's simulations for the northern Tibetan Plateau indicate a significant decreasing trend in precipitation under full-forcing conditions. However, when excluding the impact of ozone and aerosol forcing, the model delivers an increasing trend in precipitation (Supplementary Fig. 4), highlighting the critical role of aerosols in reducing precipitation levels in this region. Furthermore, multi-model ensemble mean precipitation changes over northern Tibetan Plateau from the Detection and Attribution Model Intercomparison Project (DAMIP) from the Coupled Model Intercomparison Project Phase 6 (CMIP6)[33] also confirm that this precipitation decline is consistent with being induced by anthropogenic aerosol emissions (Supplementary Fig. 4). Additionally, anthropogenic aerosols were identified as the primary driver of the weakening of the Eurasian subtropical westerly jet[34]. We therefore hypothesize, that the combined weakening of both, the AM and Westerlies[26,34–36] has led to the severe arid conditions in our study region during the 20th century. The recent drought conditions observed in central Europe and North America[37,38] suggest that this may be part of wider hemispheric event, emphasizing the global reach of these teleconnections and the extensive influence of human-induced changes on the climate system.

## Precipitation reconstruction across the Tibetan Plateau reveals coherent variations between the AM and Westerlies

The northeastern part of the Tibetan Plateau is a region that is significantly influenced by both the AM and the Westerlies, as evidenced by modern wind field maps[39] (Supplementary Fig. 5) and paleoclimate reconstructions[9,10,40]. However, the dynamics of how the AM and Westerlies interact across different timescales remain poorly understood. In this study, coherent variations between the AM and the mid-latitude Westerlies were revealed by comparing various hydroclimate proxies from regions influenced by both large-scale atmospheric circulation systems with our reconstructed precipitation time series.

On an annual scale, our precipitation reconstruction exhibits a strong correlation with the local drought reconstruction derived from the Monsoon Asia Drought Atlas[41] (MADA) within our study region, with a correlation coefficient of 0.46 for the past 706 years ($p < 0.0001$), validating the reliability of our reconstruction on an annual scale. Additionally, nearby tree-ring $\delta^{18}O$ records from the southern Tibetan Plateau[42], the western marginal region of the AM[43,44], and the Westerly region[45] (Supplementary Fig. 6) exhibit significant correlations with our precipitation reconstruction. The widespread and temporally consistent nature of these trends lend further confidence in these reconstructions, highlighting the reliability across different regions influenced by the AM and Westerly systems.

On longer timescales, spanning decades to centuries, our precipitation time series highlights a distinctive wet period from *ca.* 700–100 BC, followed by a gradual shift towards arid conditions that lasted approximately 600 years during AD 220–800. Afterwards, precipitation mainly fluctuated below the long-term average. These centennial-scale variations observed in our study align closely with other proxies representing hydroclimate conditions across monsoonal Asia[27,46,47] (Supplementary Fig. 7a–c) and Westerly-dominated central Asia[25] (Supplementary Fig. 7d). This alignment suggests a cross-archive, coherent hydroclimate signal across a broad region, underscoring the interconnectivity of climate variability and dynamics across Asia.

The consistent patterns observed from annual to millennial scales in our precipitation reconstruction and other hydroclimatic series (Supplementary Figs. 6 and 7) suggest that it not only reflects the influence of the AM (Supplementary Figs. 6a–d and 7a–c), but our reconstruction also effectively captures the impact of the Westerly circulation system (Supplementary Figs. 6e and 7d). Our reconstruction, which encompasses both high- and low-frequency hydroclimate variability, can therefore act as a pivotal link connecting the Westerlies and AM. By incorporating the effects of both climatic systems, our study provides valuable insights into the complex interaction between the Westerlies and AM, highlighting the integrated nature of global climate systems and their role in shaping regional hydroclimate patterns over extended temporal scales.

Minor dissimilarities between our precipitation reconstruction and other hydroclimate series likely originate from differences in regional hydroclimate characteristics and seasonal patterns. For example, the rainy season in tropical–subtropical South China begins earlier and lasts longer compared to the monsoon season affecting the Tibetan Plateau. These variations might also be due to the lower resolution and dating uncertainties associated with other archives (Supplementary Fig. 7d). These uncertainties often stem from changes in carbon reservoir age in lacustrine sediments over time[25,48]. With its high temporal resolution and robust dating accuracy, our precipitation reconstruction can therefore serve as a comprehensive benchmark for hydroclimate variations over the past 3.5 millennia on the Tibetan Plateau.

## Hydroclimate and development of the Chinese civilization

Early societies, particularly those relying on rain-fed agriculture in arid and semi-arid regions, were highly vulnerable to changes in hydroclimate[49,50]. A humid climate could enhance agricultural productivity, thereby supporting the spatial advancement of civilization. Conversely, droughts were frequent catalysts for famines, plague, social conflicts, and wars, often precipitating changes in dynasties[50,51]. The ancient Chinese civilization primarily flourished in central and eastern China, encompassing regions such as the Loess Plateau, the North China Plain, and the middle and lower reaches of the Yangtze River. Although our sampling sites do not cover those specific areas, hydroclimatic linkages have been found between Tibetan Plateau and central and eastern China at various timescales[52,53]. Spatial correlations show significant correlations of our reconstructed precipitation with regions in central-eastern China, especially Xi'an in Shaanxi province and Central Plains where early Chinese cultural activities occurred (Supplementary Fig. 8). The CESM-LME model was utilized to demonstrate the spatial consistency in precipitation across the northeastern of the Tibetan Plateau, the Indian subcontinent, and eastern China[6]. Therefore, our long-term hydroclimate reconstruction offers valuable insights into the relationship between climate variability and societal advancement.

The pronounced long-term wet period identified in our reconstruction from the 720 s BC to the 110 BC corresponds to the late Zhou period spanning the early Han Dynasties in China (Fig. 3d). This era was marked by significant cultural developments in ancient China, fostering an environment conducive to intellectual and philosophical progress[54]. This alignment suggests a potential link between favorable hydroclimatic conditions and the enhancement of societal condition[50,51], underscoring the impact of environmental factors on the developmental trajectory of Chinese civilization.

Our hydroclimate reconstruction indicates that the rise and fall of several Chinese dynasties corresponds with the timing of significant shifts towards arid conditions, with three distinct phases of long-term precipitation decline following the long humid period (Fig. 3d). The first drought phase from 110 BC to AD 280 corresponds well with the rise and fall of the Han Dynasty and Wang Mang's Xin Dynasty[55]. The second phase (AD 330–770) aligns with the rule of the local Tibetan

Tuyuhun Tribe during the Era of Disunity[56], Sui and Tang Dynasties[55]. The third phase (AD 950–1300) corresponds with the Song Dynasty[55]. The humid conditions in the earlier stages of each dynasty were associated with periods of prosperity, while the progressive shift toward aridity consistently coincided with their decline and eventual collapse.

During the first drought phase (110 BC–AD 280), the short-lived Xin Dynasty (AD 9–23) was established and then rapidly declined. Initially, the climate was humid, but it quickly became arid after AD 14, leading to a significant increase in famine victims and instances of cannibalism in the capital Chang'An (now Xi'an)[57]. This contributed to widespread uprisings and civil unrest, culminating in the dynasty's overthrow in AD 23. The droughts continued until AD 24, which recorded only 151.4 mm of precipitation, 2.8 σ below the average (27.6% of the mean), marking it as the eighth driest year in the past 3476 years (Fig. 3e). Another severe drought occurred between AD 220–280 (Fig. 3d), during which precipitation levels sharply fell by 3 σ from the most humid period around 130 BC. The extreme drought of AD 243, with precipitation of only 132.2 mm (3.3 σ below average), was the second driest in the past 3500 years. This period coincided with the Three Kingdoms Period (AD 220–280; Fig. 3d) wars, which, along with the widespread famine caused by the drought, led to a dramatic decrease in China's population from 60 million to 30 million[58].

The Qin Dynasty, founded in 221 BC as China's first unified and centralized government, experienced humid hydroclimate conditions[59] (Fig. 3d). While these conditions likely contributed to the dynasty's rise, they also contributed to its downfall through excessive rainfall and floods, which catalyzed a peasant revolt in July 209 BC[60]. This scenario is notably different from other Chinese dynasties, whose demise was often triggered by persistent or severe droughts.

Our 3476-yr tree-ring $\delta^{18}O$-based precipitation reconstruction from the northeastern part of the Tibetan Plateau not only documents historical annual fluctuations of precipitation but also encompasses remarkable long-term hydroclimatic trends. It unveils a distinct wet period during 720–110 BC, which fostered the societal prosperity and progress of the late Zhou and early Han Dynasties. Conversely, droughts typically coincided with social unrest and crises, and even the collapse of dynasties. This long-term precipitation reconstruction also enables us to identify the period of AD 1930 to 2010 as the driest in the last 3476 years at a centennial scale. This drought trend is likely impacted by weakened atmospheric circulations, attributed in part to increased anthropogenic aerosol emissions in the 20th century. Although contemporary society has developed enhanced resilience against drought compared to ancient eras, the potential for more frequent, severe, and prolonged droughts in the future poses an increasing threat to ecosystems and human civilization. This emphasizes the urgency of substantial emissions reductions and nature conservation efforts. Our results introduce a hydroclimate perspective on the region's climate history and dynamics based on long records, offering climate modelers, archaeologists, and historians an invaluable lens to explore past events in greater depth.

## Methods

### Tree-ring cellulose extraction and $\delta^{18}O$ measurement

Annual rings were separated under a microscope and transferred into labeled glass tubes. The Jayme-Wise method was employed to extract α-cellulose[61]. Samples were treated with a toluene-ethanol mixture (1:1) at 60 °C for 1 h, repeated three times. They were then soaked in acetone and incubated in a 60 °C water bath for 1 h. Subsequently, samples were exposed to a NaClO$_2$-acetic acid solution at 80 °C for 1 h, repeated three times. Next, 17.5% NaOH was added, and the samples were heated at 80 °C for 45 min, also repeated three times. The α-cellulose was washed until a neutral pH was achieved. Finally, ultrasonic homogenization was applied to efficiently decompose and blend cellulose fibers[62], followed by freeze-drying.

The resulting homogenized cellulose samples were subsequently packed into silver capsules and pyrolyzed to CO in an elemental analyzer (TC/EA, Thermo Fisher, Germany) coupled to an Isotope Ratio Mass Spectrometer (Delta V Advantage, Thermo Fisher, Germany). The resulting $\delta^{18}O$ values represent the permil deviation of the $^{18}O/^{16}O$ ratio of the cellulose sample from the given $^{18}O/^{16}O$ ratio of the Vienna Standard Mean Ocean Water (VSMOW). Sample values were calibrated against laboratory standard cellulose (cellulose microcrystalline, Merck, Germany), and against the IAEA-601 standard (reference material of International Atomic Energy Agency), inserted every eight samples during the measurement process. Standard measurements result in an overall analytical precision for the tree-ring $\delta^{18}O$ analysis of ≤0.2‰.

### Offset-detection among individual tree-ring $\delta^{18}O$ series

We identified the out-of-range offset trees as those whose mean exceeded the rolling mean of the same period plus or minus the standard deviation of the calibration period[19]. Trees with a mean value outside of this range are designated as outliers and would require an interactive adjustment process, but no outlier trees were detected in our study.

### EPS and Rbar

The EPS was used to evaluate the statistical agreement between the $\delta^{18}O$ series (or the common variance relative to the total variance). Generally, an EPS value greater than 0.85 is a recommended threshold for a reliable chronology. The Rbar parameter indicates the average of correlations of all pairs of $\delta^{18}O$ series. In this study, EPS and Rbar were calculated over 50-year windows, lagged by 25 years, to evaluate the reliability of our tree-ring $\delta^{18}O$ chronology[63]. The mean EPS value was 0.90, which was considerably larger than the recommended threshold of 0.85. It should be noted that four 50-year-long periods, centered at 436 BC (EPS = 0.84), AD 715 (EPS = 0.76), AD 815 (EPS = 0.82), and AD 1715 (EPS = 0.84), exhibit EPS values below the threshold of 0.85.

### Tibetan Plateau tree-ring $\delta^{18}O$ climate response analysis and 3476-yr annual precipitation reconstruction

Climate data from ten meteorological stations (Fig. 1b) with relatively long records near the sampling sites were utilized for the climate response analysis. Despite spatial heterogeneity in precipitation in this topographically complex plateau, site records reveal similar monthly and annual variability. Pearson correlation coefficients between the tree-ring $\delta^{18}O$ chronology and climate data from these ten stations (Fig. 1b, Supplementary Fig. 1) also show similar response patterns (Supplementary Fig. 2). Thus, a regional climate mean was generated using the scaling method described by Yang et al.[6] The climate time series at each individual station in each month of the year were scaled by linear transformations. After the scaling process, the climate data at each station share the same average and variance as the rough composite data in the corresponding month during the common period (AD 1960–2010). Finally, the monthly scaled data from the ten stations were arithmetically averaged to produce the regionally scaled climate series. Proxy-climate correlation analysis was performed against monthly and seasonal scaled precipitation, temperature, and relative humidity records from July of the previous year to October of the current year of tree-ring formation for each station and for the regional records to identify the climate response pattern of the Tibetan Plateau tree-ring $\delta^{18}O$ during the calibration period spanning from AD 1960 to 2010.

The Tibetan Plateau tree-ring $\delta^{18}O$ chronology was significantly negatively correlated with precipitation and relative humidity conditions during previous August–October and current May–August, whereas no significant relationship was found between our tree-ring $\delta^{18}O$ chronology and temperature (Fig. 3a). Compared to relative humidity, precipitation revealed a stronger association with the

tree-ring $\delta^{18}O$ series. After combining the monthly dataset, tree-ring $\delta^{18}O$ had the strongest significant relationship with the regional precipitation amount from previous September to current August ($P_{S-A}$), with a correlation coefficient of −0.78 ($p < 0.0001$). Consequently, $P_{S-A}$ was reconstructed with the following regression function:

$$P_{S-A} = -15.546 \times \delta^{18}O + 724.336 \quad (1)$$

($n = 50$, $r = -0.78$, $R^2 = 0.606$, $R^2_{adj} = 0.598$, $F = 73.975$, $p < 0.0001$, Durbin-Watson value = 2.015)

### Split calibration-verification method

The split calibration-verification method[63] was used to evaluate reliability and stability of the regression function (Eq. 1) during the calibration and verification stages. These validation trials were performed by calibrating the Tibetan Plateau meteorological precipitation data for one subperiod (AD 1961–1990 and AD 1981–2010) and validating the reconstruction model with the data remaining from the other period (AD 1991–2010 and AD 1961–1980, respectively). The quality of our results was evaluated by the correlation coefficient ($r$), the sign test ($ST$), the reduction of error test ($RE$), the coefficient of efficiency ($CE$), and the product means test ($t$) during the verification period. Generally, in tree-ring studies, $RE$ and $CE$ values greater than zero indicate rigorous and robust model skill[18,63], and additionally higher values of $RE$ and $CE$ indicate better results. Moreover, the values of $CE$ are more rigorous and are typically lower than those of $RE$ (Supplementary Table 4).

### Definition of drought periods

Drought periods were defined as intervals of more than one standard deviation (1σ) below the mean of entire reconstruction. Although the precipitation during AD 220–280 (Fig. 3d) was not lower than mean−1σ, it fell by 3σ rapidly from the most humid period (*ca*. 130 BC). The resulting significant reduction in precipitation corresponds with a famous drought event that occurred during the "Three Kingdom War".

### Effective number of degrees of freedom estimation

Due to autocorrelations in the data used in this study and their corresponding reduction in number of degrees of freedom, the effective number of degrees of freedom (EDOF) was estimated to test the significance level of correlations for each pair of time series[64]. The EDOF was estimated by:

$$EDOF = \frac{2 \times \Delta T}{T_{c1}}(N - 2) \quad (2)$$

where $N$ denotes the length of the time series, $\Delta T$ and $T_{c1}$ refer to the sampling interval and the cutoff period in the low-pass filtering, respectively.

### Interpolation approach and smoothing method for stalagmites and lacustrine records

Samples of stalagmites and lacustrine deposits are generally treated according to the depth/length of the material and only dated at certain positions, so the age of each sample is obtained by interpolation between a few data points, and the time intervals of individual data points of the entire series are generally uneven. To ensure valid and meaningful correlations between series derived from different terrestrial proxy archives (stalagmites, and lacustrine deposits) and our Tibetan Plateau tree-ring $\delta^{18}O$ record, we first interpolated the terrestrial proxy series into the corresponding time axis of our tree-ring chronology. Then, both the tree-ring $\delta^{18}O$ record and stalagmite/lacustrine series were low-pass filtered at various time scales[65]. Lastly, we calculated the correlation coefficients between our tree-ring $\delta^{18}O$

record and the other low-pass filtered series, and their significance levels were determined under the EDOF reduction.

## Data availability

The reconstructed 3500-year precipitation on the Tibetan Plateau can be found at http://paleodata.ieecas.cn/FrmDataInfo_EN.aspx?id=3ba7d8ee-3878-4cfa-a652-479f2226db73 and are freely available at the NOAA Paleoclimatology Database. The data of the ETOPO1 Global Digital Elevation Model can be found at https://www.ngdc.noaa.gov/mgg/global/relief/ETOPO1/tiled/. Source data are provided with this paper.

## Code availability

The code used in this study can be found on GitHub at https://github.com/TroicyFun/TPP.

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

## Acknowledgements

We thank Professor J. Dodson, G. H. Schleser for their generous help. Y.L. acknowledges grants from the Second Tibetan Plateau Scientific Expedition and Research (2019QZKK0101), the Chinese Academy of

Sciences (XDB40010300), the National Natural Science Foundation of China (42361144712 and U1803245), the Fund of Shandong Province (LSKJ202203300), the National Observation and Research Station of Regional Ecological Environment Change and Comprehensive Management in the Guanzhong Plain, Shaanxi, the IEECAS and the SKLLQG; C.M.Z. acknowledges support from the SNSF Ambizione program (#PZ00P3_193646). This is a contribution number 202401 of SISTRR.

## Author contributions

Conceptualization: Y.L., H.S., and Q.L.; Methodology: Y.L., Q.L., H.S., C.F., Q.C., C.S., R.L., M.R., L.W., and L.C.; Visualization: H.S., Y.S., C.F., Q.Z., and C.S.; Funding acquisition: Y.L. and C.M.Z.; Investigations: all authors; Project administration: Y.L.; Supervision: Y.L. and Z.A.; Writing–original draft: Y.L., H.S., Q.L., S.W.L.; C.M.Z., and L.M.; Writing–review & editing: T.W.C., C.M.Z., Y.L., H.S., S.W.L., Q.C., L.M., U.B., W.C., Q.L., D.C., H.W.L., A.B., K.T., J.G., W.Z., A.S., H.C., Y. Lei, J.S., W.G., L.N., L.W. and X.L..

## Competing interests

The authors declare no competing interests.

## Additional information

[1]The State Key Laboratory of Loess and Quaternary Geology, Institute of Earth Environment, Chinese Academy of Sciences, Xi'an, China. [2]CAS Center for Excellence in Quaternary Science and Global Change, Chinese Academy of Sciences, Xi'an, China. [3]Institute of Global Environmental Change, Xi'an Jiaotong University, Xi'an, China. [4]The Laboratory of Tree-Ring Research, The University of Arizona, Tucson, AZ, USA. [5]Department of Geography, University of Cambridge, Cambridge, UK. [6]Global Change Research Institute (CzechGlobe), Czech Academy of Sciences, Brno, Czechia. [7]Department of Geography, Faculty of Science, Masaryk University, Brno, Czechia. [8]Institute of Mountain Hazards and Environment, Chinese Academy of Sciences, Chengdu, China. [9]Research Unit Forest Dynamics, Swiss Federal Research Institute (WSL), Birmensdorf, Switzerland. [10]Oeschger Centre for Climate Change Research, University of Bern, Bern, Switzerland. [11]Xi'an Institute for Innovative Earth Environment Research, Xi'an, China. [12]Institute of Integrative Biology, ETH Zurich (Swiss Federal Institute of Technology), Zurich, Switzerland. [13]Centre for Southern Hemisphere Ocean Research (CSHOR), CSIRO Oceans and Atmosphere, Hobart, TAS, Australia. [14]Institute of Geography, Friedrich-Alexander-University Erlangen-Nürnberg, Erlangen, Germany. [15]Department of Environment and Biodiversity, University of Salzburg, Salzburg, Austria. [16]Department of Earth Sciences, University of Gothenburg, Gothenburg, Sweden. [17]Department of Earth Science, California State University, Dominguez Hills, Carson, CA, USA. [18]Institute of Subtropical Agriculture, Chinese Academy of Sciences, Changsha, China. [19]School of Archaeology and Museology, Peking University, Beijing, China. [20]College of Atmospheric Sciences, Chengdu University of Information Technology, Chengdu, China. [21]Key Laboratory for Virtual Geographic Environment, Ministry of Education, School of Geography, Nanjing Normal University, Nanjing, China. [22]State Key Laboratory Cultivation Base of Geographical Environment Evolution of Jiangsu Province, School of Geography, Nanjing Normal University, Nanjing, China. [23]Jiangsu Center for Collaborative Innovation in Geographical Information Resource Development and Application, School of Geography, Nanjing Normal University, Nanjing, China. [24]Institute for Advanced Ocean Study (IAOS), Ocean University of China, Qingdao, China. ✉e-mail: liuyu@loess.llqg.ac.cn; songhm@xjtu.edu.cn

