## [Peer Review file · Nature Communications]

Recent centennial drought on the Tibetan Plateau is outstanding within the past 3500 years

Corresponding Author: Professor Yu Liu

Version 0:

Reviewer comments:

Reviewer #1

(Remarks to the Author)

This manuscript presents a precisely-dated 3476-year precipitation reconstruction derived from tree-ring oxygen data on the Tibetan Plateau. This is the longest annually-resolved terrestrial oxygen record to date. Three main conclusions are drawn. However, data analysis of this study is from my perspective not valid to allow drawing such general conclusions.

One of the most important findings of this manuscript is that the 20th century drought extremes are unparalleled within the past three millennia. Unfortunately, this conclusion is untenable for these reasons.

The oxygen isotopic values in precipitation varies greatly with elevation, i.e., altitude effect. Thus, one has to be very careful when constructing a homogeneous oxygen isotope sequence. It is noted that the living trees that were used during the calibration period originate from one site, SG. One question arises how well trees from other sites with different elevations used in earlier parts of the chronology represent the climate signal derived in the calibration period. As a result, the mean of all the individual isotope records is mandatory for estimation of the temporal robustness and representativeness of the chronology.

It is noted that the tree samples consist of different sources, including archaeological specimens discovered in old tombs, dead and living trees. These samples were collected from four sites, DGM, XTT, SG, and DL, with a large elevation gradient ranging from 2900-4100 m. Figure 1 shows 4 tree sites, some of which are 200 kilometers apart. These large horizontal and vertical distances definitely result in large differences in the oxygen isotope values of tree rings over time.

Age-related trends in individual stable oxygen timeseries have not been removed.

The second point that the authors applauded is that the drought in the 20th century is linked to the weakening of both the AM and westerlies due to anthropogenic aerosol emissions. This is equally unconvincing. Robust evidence needs to be provided (see my detailed comments below).

The third point that they highlighted is that three identified stages are characterized by shifts toward arid hydroclimate conditions, which corresponded to the significant social unrest and dynasties collapse. Unfortunately, social unrest and dynasties alterations are mainly represented in eastern China rather than in western China or the NE Tibetan Plateau. One first needs to confirm that the oxygen isotope sequence is representative of climate change in eastern China. Unfortunately, the key step is missing.

Lines 118-122: The argument is problematic. It is true that tree-ring oxygen record can reflect large-scale atmospheric circulation patterns in tropical areas. However, it is questionable in this study region due to strong evaporative leaf water enrichment. As the authors wrote: Decreases in precipitation, typically associated with increasing aridity, lead to higher $\delta^{18}O$ enrichment in leaf and soil water due to elevated atmospheric vapor pressure deficit.

Lines 154-157: the citations that shows carbohydrates formed in the previous year substantially contribute to earlywood formation are not necessarily applicable to the Tibetan Plateau.

Lines 200-203: the authors use tree-ring-width data from north-central China, stalagmite records from the southeastern Tibetan Plateau and lake sediment records from central Asia to verify the extreme drought intensity in the 20th century. Such comparisons are inadvisable because precipitation variations differ regionally.

Lines 208-223: the authors used the simulation data from Community Earth System Model Last Millennium Ensemble (CESM-LME) project to demonstrate the 20th century decreasing trend in precipitation. One wonders if the model performs well in this high-elevation area of unusually complex terrain. In addition, it is strange how the authors concluded that the combined weakening of both the Asian Monsoon and Westerlies has led to this severe aridity in China during the 20th century. It is also inappropriate to overemphasize human activity (i.e., increased anthropogenic aerosol emissions) impact on the climate system.

Lines 234-235: the authors used the word 'local'. Actually, the Monsoon Asia Drought Atlas 235 (MADA) is not local but regional or at a much larger spatial scale.

Lines 234-252: these comparisons mean not much to me since the correlation is so weak with maximum explained variance being 0.2-0.3. Moreover, it should be noted that these low-resolution proxy records such as stalagmites, lake sediments, and ice cores have large dating uncertainties. Correlation does not mean causation, especially when only a small fraction of the available data is shown. I am concerned HOW the fraction of globally distributed records were chosen considering a large number of proxy records are available. As the authors wrote: These variations might also be due to the lower resolution and dating uncertainties associated with other archives. These uncertainties often stem from changes in carbon reservoir age in lacustrine sediments over time.

Lines 282-288: It is hard to understand WHY and HOW the wet period identified in precipitation reconstruction from the 720s BC to the AD 70s could be linked to significant cultural developments-intellectual and philosophical advancements in ancient China. Correspondence does not imply causation. The same problem is with the lines 289-298, 299-312 and 313-318. When linking the rise and fall of dynasties to climate system, one must be aware of the spatial coherence of climate change.

Reviewer #2

(Remarks to the Author)

The hydroclimate of the Tibetan Plateau is strongly influenced by both the presence of the Asian Monsoon and Westerlies. In recent years, there have been several pioneering papers reconstructing the climate of the region with the primary focus upon Asian monsoonal precipitation. Many of the co-authors of this paper are linked to this independent research. For example, Yang et al. (2021; not cited) published a 6,700-year isotope chronology from Delingha that represented the "longest existing precisely dated isotope chronology in Asia" typically with a one-to-five-year resolution. The novelty of THIS STUDY is that it presents the longest precipitation reconstruction from a 3476-year precisely dated annually resolved oxygen isotope chronology. The reconstruction allows current droughts to be investigated against natural climate variability over three millennia. It's a word that gets used to often these days, but these droughts appear to be "unprecedented." However, the time-series end in the year AD 2010 illustrating the need to continual chronology development.

The links to societal changes are speculative in places but cannot be discounted. Interpretation should be cautious especially when sample replication is low. Overall, the paper makes a significant contribution towards the field. I recommend publication subject to minor changes.

L78. As the time-series end in AD2010, the authors should refer to 20th (and 21st century!) droughts.

L83-84. This comment appeared non-sequitur, but it is partly justified later in the text (L325-335). It is acceptable.

L135-144. Although it's not totally prescriptive, what was the typical EPS value found in THIS STUDY to constitute a robust dataset? This is important. It is also not covered in the Methods (L.473-478) and too difficult to read on Figure 2 (L602).

L144 (and later). It should be emphasised that any interpretation covering the first six hundred years of the time-series should be cautious as it is based upon a single tree. This is not uncommon but it should be acknowledged.

L193 (and later). This period also includes ten-years of the 21st century.

L462. Please provided supporting evidence for the homogenisation process. I presume that it is based upon Laumer et al. (2009).

L499-502. What is the possible cause of the observed relationship?

L515 (& EXTENDED L156). The periods used for the split calibration-verification are not truly independent (AD 1961–1990 and AD 1981–2010). Is this approach acceptable?

L590. Figure 1a. Carefully re-select the colors as yellow font is difficult to read on a colored background.

L602. Figure 2. Strictly there is no zero in the AD/BC timescale as AD refers to in the “year of Our Lord” so 1BC precedes AD1. Many scientists now use the on-secular Common Era timescale (BCE/CE) but the same issue remains. The approach adopted here has been used elsewhere, but how do the authors justify the use of zero as it doesn't exist.

L602. Clarity required with the EPS and Rbar values. At how many points is EPS less than the generally accepted value of 0.85?

EXTENDED L135. Sample XTT is from a lower elevation. Will it still respond to the same drivers?

EXTENDED L155. The period covered should be AD 1961-2010.

Recommended additional references:

Laumer, W., et al. (2009). "A novel approach for the homogenization of cellulose to use micro-amounts for stable isotope analyses." *Rapid Communications in Mass Spectrometry* 23(13): 1934-1940.

Yang, B., et al. (2021). "Long-term decrease in Asian monsoon rainfall and abrupt climate change events over the past 6,700 years." *Proceedings of the National Academy of Sciences* 118(30): e2102007118.

Version 1:

Reviewer comments:

Reviewer #1

(Remarks to the Author)

I appreciate Authors' effort to improve the original submission during the revision stage. However, I do see some flaws that would prohibit publication in its current form.

One of the most important conclusions of this manuscript is that recent drought on the Tibetan Plateau is unprecedented in the past 3500 years. It's crucial to ensure that such statements are supported by robust evidence. The title should reflect the findings accurately and not overstate them based on the current data. What is meant by the word 'recent'?

In general, the oxygen isotopic ratio varies greatly with elevation, i.e., altitude effect. I would suggest the authors exemplify how the isotope mean of each sample varies over time and with elevation. In addition, an altitude-isotope relationship should be plotted to visually inspect if the altitude effect exists or not.

The "outliers correction method" doesn't seem to answer my concerns. The authors stated that they found no outlier trees in their study, suggesting no significant offset among the 17 $\delta^{18}\text{O}$ tree-ring series that were analyzed. That being the case, how can you come to that conclusion "Recent drought on the Tibetan Plateau is unprecedented in the past 3500 years"?

The provenance and growing elevation of the XTT and DGM archaeological wood, covering the period from 1465 BC to 86 BC, are important aspects that need clarification. From Supplementary Fig. 8, the overall $\delta^{18}\text{O}$ series encompassing all samples are notably positively biased than the tree-ring $\delta^{18}\text{O}$ chronology at the SG site. It will inevitably lead to biases and non-homogeneity of the final chronology. I agree with the second reviewer on the comment 'any interpretation covering the first six hundred years of the time-series should be cautious as it is based upon a single tree'. It can barely be used to past-present comparison. Taking all these points into consideration, the credible time period for the chronology seems to be limited to the last 2000 years.

The second reviewer mentioned that Yang et al. (2021) published a 6,700-year isotope chronology from Delingha that represented the "longest existing precisely dated isotope chronology in Asia" typically with a one-to-five-year resolution. Considering the dataset is from the same study area, it is essential and beneficial to add a figure showing how different the two chronologies are.

The authors used CESM-LME and CMIP6 multi-model aerosol-only forcing simulation to demonstrate the 20th century decreasing trend in precipitation. However, it is well accepted that the climate was warm-dry and then changed to warm and wet from the end of the 19th century to the 1970s. Obviously, the model performance is flawed in simulating precipitation variations.

Lastly, it is absolutely essential that the authors share their isotopic measurement data, preferably on a modern platform like GitHub or at Zenodo since this is a routine exercise in transparency and reproducibility.

Reviewer #2

(Remarks to the Author)

The authors have address my minor concerns with the manuscript with the exception of one issue. They still include a "zero year" on the AD/BC timescale yet this year never actually existed. They quote a paper in support of this approach (Buentgen and Oppenheimer, 2020) but it is controversial. Many other established long chronologies are reported without this error. To avoid ambiguity, they should correct this error or state exactly what has been done.

I will let the editor decide upon this minor concern. Otherwise, I happily recommend publication by Nature Communications.

Version 2:

Reviewer comments:

Reviewer #1

(Remarks to the Author)

Thank you for addressing my concerns, and the manuscript is much improved. Visual inspection finds the recent drought does not exceed the 95% confidence range of natural climate variability, indicating that it is not unprecedented in terms of statistics (Fig.3). The use of this word unprecedented in title and relevant text is overstated.

Figure R1 Visualization of the relationship between average tree-ring $\delta^{18}\text{O}$ values from this study with altitude is important, and it would be helpful if it is added in the supplement.

Version 3:

Reviewer comments:

Reviewer #1

(Remarks to the Author)

I am pleased the authors have answered my concerns. The manuscript is well-structured now, and can be accepted.

Reply to REVIEWER COMMENTS

Reviewer #1 (Remarks to the Author)

This manuscript presents a precisely-dated 3476-year precipitation reconstruction derived from tree-ring oxygen data on the Tibetan Plateau. This is the longest annually-resolved terrestrial oxygen record to date. Three main conclusions are drawn. However, data analysis of this study is from my perspective not valid to allow drawing such general conclusions.

[Response]

Thank you for your valuable comments. We sincerely appreciate your recognition of our chronology as the longest tree-ring $\delta^{18}\text{O}$ chronology with annual resolution on the Tibetan Plateau. The authors involved in this manuscript bring expertise from diverse fields such as dendrochronology, geochemistry, statistical analysis, climate modeling, and historical climatology. This article has undergone extensive discussions, calculations, revisions and iterative refinements among all co-authors over an extended period of time. We hope that our responses and the resulting revised version of our manuscript addresses your concerns satisfactorily.

One of the most important findings of this manuscript is that the 20th century drought extremes are unparalleled within the past three millennia. Unfortunately, this conclusion is untenable for these reasons. The oxygen isotopic values in precipitation varies greatly with elevation, i.e., altitude effect. Thus, one has to be very careful when constructing a homogeneous oxygen isotope sequence.

[Response]

Thank you for your comments. We acknowledge and agree that in general altitude can alter the oxygen isotope composition of the source water signal, i.e. precipitation. However, in using the "outliers correction method" from Arosio et al. (2024) and McCarroll et al. (2004), we found no outlier trees in our study, suggesting no significant offset among the 17 $\delta^{18}\text{O}$ tree-ring series that were analyzed in this study. Additionally, previous investigations have shown no altitude-related effects on tree-ring $\delta^{18}\text{O}$ values on the northern Tibetan Plateau (Xu et al., 2017; Yang et al., 2021). We will provide further supporting evidence in the following sections.

References:

- Arosio, T., et al., 2024. Methodological constrains of tree-ring stable isotope chronologies. *Quaternary Science Reviews*, 340, 108861.
- McCarroll, D., Loader, N.J., 2004. Stable isotopes in tree rings. *Quaternary Science Reviews*, 23, 771–801.
- Xu, C.X., et al., 2017. Negligible local-factor influences on tree ring cellulose $\delta^{18}\text{O}$ of Qilian juniper in the Animaqing Mountains of the eastern Tibetan Plateau. *Tellus Series B-Chemical and Physical Meteorology*, 69, 1391663.
- Yang, B., et al., 2021. Long-term decrease in Asian monsoon rainfall and abrupt climate change events over the past 6,700 years. *Proceedings of the National Academy of*

It is noted that the living trees that were used during the calibration period originate from one site, SG. One question arises how well trees from other sites with different elevations used in earlier parts of the chronology represent the climate signal derived in the calibration period. As a result, the mean of all the individual isotope records is mandatory for estimation of the temporal robustness and representativeness of the chronology. It is noted that the tree samples consist of different sources, including archaeological specimens discovered in old tombs, dead and living trees. These samples were collected from four sites, DGM, XTT, SG, and DL, with a large elevation gradient ranging from 2900–4100 m. Figure 1 shows 4 tree sites, some of which are 200 kilometers apart. These large horizontal and vertical distances definitely result in large differences in the oxygen isotope values of tree rings over time.

[Response]

The primary concern raised by the reviewer is the feasibility of averaging tree-ring isotopes from different locations. We appreciate your valuable comments, as this issue is crucial for the reliability of tree-ring isotope reconstructions. To address this, we sought to identify outlier trees based on whether their mean values fell outside the running mean of the same period, plus or minus the standard deviation of the calibration period (Arosio et al., 2024). Our analysis found no outlier trees in our study.

In designing our research plan, we carefully considered sample selection. Our dataset comprises 17 samples, including 12 from living and dead trees at the SG site, spanning from 700 BC to AD 2010, and one sample from the DL site, covering AD 1082–1974. To extend the chronology, we also included four samples from ancient tombs, a common approach in tree-ring research (Büntgen et al., 2021; Nakatsuka et al., 2020; Naulier et al., 2015). Therefore, the SG samples constitute the primary dataset for this chronology. If the DL and archaeological samples are excluded, the remaining chronology from the SG site samples alone is highly correlated with the complete dataset ($r=0.975$, 760 BC–AD 2010) (Figure R1). In addition, despite a maximum 200 km distance between our sample locations, no significant discrepancies in terms of different levels in $\delta^{18}\text{O}$ values were observed. Therefore, we retained the DL sequence in our overall chronology.

Although archaeological wood (Figure R2) was sourced from ancient tombs at lower elevations, the juniper forests from which it came grow on slopes with elevations between 3500–4100 meters (Shao et al., 2010) (Figure R3). As a result, this wood likely shares similar growth characteristics compared to the living and dead tree samples from the SG site. Additionally, elevation variations among SG samples are minimal (around 300 meters), and the resulting highly correlated $\delta^{18}\text{O}$ series among trees within the SG site clearly indicate that any elevation effect on $\delta^{18}\text{O}$ values is negligible.

The similarity in tree-ring $\delta^{18}\text{O}$ values across different sites can be attributed to the uniform characteristics of $\delta^{18}\text{O}$ and the topographically and climatologically similar terrain surrounding the Qaidam Basin. Unlike tree-ring width, which can vary locally, $\delta^{18}\text{O}$ values reflect broader atmospheric circulation patterns (Büntgen et al., 2021; Li et

al., 2020; Nakatsuka et al., 2020). Despite the overall complexity of the Tibetan Plateau, the topography of the northeastern edge of the Qaidam Basin is relatively uniform over long distances with no significant changes (Figure R4). Given this, study sites 200 km apart are ecoclimatologically similar in the context of our study region's size. Therefore, similar elevation and terrain characteristics contribute to observed consistent $\delta^{18}\text{O}$ values across samples and sites. Consequently, we can postulate that latitude and distance do not significantly affect the $\delta^{18}\text{O}$ values of trees from various locations in this study.

We have added more evidence in the new version, please see line 149–154, 385–390 in the maintext and Supplementary note, as well as Supplementary Fig. 8.

Figure R1. Comparison between the tree-ring $\delta^{18}\text{O}$ chronology and $\delta^{18}\text{O}$ levels derived from living and dead trees at the SG site (blue line), and the overall mean site series encompassing all samples (gray line).

This figure has been added as Supplementary Fig. 8 in the Supplementary information.

Figure R2. DGM sampling site (The tombs are distributed on the piedmont alluvial fan at an elevation of approximately 2900–3000m. No recently growing juniper trees are currently observed within this altitude range. It is evident that the wood utilized in constructing the ancient tombs must originate from a nearby slope).

Figure R3. Overview on the SG sampling site, where living and dead trees spanning from 760 BC to AD 2010 were used.

Figure R4. Relief map and location of the sampling sites DGM, XTT, SG and DL in the geographical context of the Qaidam Basin.

References:

- Arosio, T., et al., 2024. Methodological constrains of tree-ring stable isotope chronologies. *Quaternary Science Reviews*, 340, 108861.
- Büntgen, U., et al., 2021. Recent European drought extremes beyond Common Era background variability. *Nature Geoscience*, 14, 190–196.
- Li, Q., et al., 2020. Delayed warming in Northeast China: Insights from an annual temperature reconstruction based on tree-ring $\delta^{18}\text{O}$. *Science of the Total Environment*, 749, 141432.
- Nakatsuka, T., et al., 2020. A 2600-year summer climate reconstruction in central Japan by integrating tree-ring stable oxygen and hydrogen isotopes. *Climate of the Past*, 16, 2153–2172.
- Naulier, M., et al., 2015. A millennial summer temperature reconstruction for northeastern Canada using oxygen isotopes in subfossil trees. *Climate of the Past*, 11, 1153–1164.
- Shao, X., et al., 2010. Climatic implications of a 3585-year tree-ring width chronology

from the northeastern Qinghai-Tibetan Plateau. *Quaternary Science Reviews*, 29, 2111–2122.

Age-related trends in individual stable oxygen timeseries have not been removed.

[Response]

In the publication by Treydte et al. (Nature, 2006) it was postulated, that a tree-ring $\delta^{18}\text{O}$ age-related bias is minor and did not change the overall centennial-scale (low-frequency) pattern of reconstructed precipitation. Other studies on stable oxygen isotope measurements also confirmed the absence of age trends in time series of tree-ring $\delta^{18}\text{O}$ (Büntgen et al., 2020; Duffy et al., 2019). Therefore, there is no need to detrend tree-ring $\delta^{18}\text{O}$ data to remove growth-related trends. Preservation of both, low-frequency and high-frequency signals underscores the effectiveness of $\delta^{18}\text{O}$ as a climate indicator compared to the tree-ring width parameter for large-scale reconstructions.

Specifically, for the Tibetan Plateau, Xu et al. (2017) compared $\delta^{18}\text{O}$ values in Qilian juniper trees (Figure R5)—similar to those used in our study—across different age groups and altitudes in the eastern plateau. Their analysis showed no significant differences in mean values and long-term trends between chronologies from old and young trees, suggesting that young trees are equally valid for paleoclimate reconstruction. A recent publication additionally indicates that tree-ring $\delta^{18}\text{O}$ chronologies from many regions in the wider Asia context, including the northeastern Tibetan Plateau, do not exhibit clear age-related trends (Xu et al., 2024).

Additionally, to address potential juvenile effects and to account for possible early growing age trends, we excluded the first 30 years from each sample, even though such effects were not observed in our study region (Yang et al., 2021).

[Redacted]

Figure R5. The comparison of tree-ring $\delta^{18}\text{O}$ values from young trees (red lines) and old trees (black lines). The dashed line indicates the pith year. (This figure was from Xu et al., 2017)

References:

- Büntgen, U., et al., 2020. No age trends in oak stable isotopes. *Paleoceanography and Paleoclimatology*, 35, 4.
- Duffy, J.E., et al., 2019. Absence of age-related trends in stable oxygen isotope ratios from oak tree rings. *Global Biogeochemical Cycles*, 33, 841–848.
- Treydte, K.S., et al., 2006. The twentieth century was the wettest period in northern Pakistan over the past millennium. *Nature*, 440, 1179–1182.
- Voelker, S.L., et al., 2018. Millennial-scale tree-ring isotope chronologies from coast redwoods provide insights on controls over California hydroclimate variability. *Oecologia*, 187, 897–909.
- Xu, C.X., et al., 2017. Negligible local-factor influences on tree ring cellulose $\delta^{18}\text{O}$ of Qilian juniper in the Animaqing Mountains of the eastern Tibetan Plateau. *Tellus Series B-Chemical and Physical Meteorology*, 69, 139166.
- Xu, C.X., et al., 2024. Tree ring oxygen isotope in Asia. *Global and Planetary Change*, 232, 104348.
- Yang, B., et al., 2021. Long-term decrease in Asian monsoon rainfall and abrupt climate change events over the past 6,700 years. *Proceedings of the National Academy of Sciences*, 118, e2102007118.

The second point that the authors applauded is that the drought in the 20th century is linked to the weakening of both the AM and westerlies due to anthropogenic aerosol emissions. This is equally unconvincing. Robust evidence needs to be provided (see my detailed comments below).

[Response]

Please see our detailed responses below.

The third point that they highlighted is that three identified stages are characterized by shifts toward arid hydroclimate conditions, which corresponded to the significant social unrest and dynasties collapse. Unfortunately, social unrest and dynasties alterations are mainly represented in eastern China rather than in western China or the NE Tibetan Plateau. One first needs to confirm that the oxygen isotope sequence is representative of climate change in eastern China. Unfortunately, the key step is missing.

[Response]

Thank you for your helpful suggestions.

It is true that most major dynasties in ancient China were concentrated in the central and eastern regions. Though the natural environment of Qaidam basin is harsh for human survival, humans began to settle here since the Bronze Age (Chen et al., 2015). Tuyuhun was a significant ancient civilization in the Qaidam Basin, which experienced its collapse during AD 663, coinciding with an arid climatic phase evident in our precipitation reconstruction. We briefly illustrate the linkage of hydroclimate variability between the northeastern Tibetan Plateau and eastern China across both, short and long timescales.

1) On inter-annual scale:

Spatial correlations showed that our Tibetan Plateau precipitation reconstruction

is significantly related to already published reconstructed precipitation records from the Tibetan Plateau and also other regions in central-eastern China, such as Xi'an in the Shaanxi Province and the Central Plains in Henan Province, where early Chinese cultural activities were prominent (Figure R6). The spatial precipitation during AD 850-1850 was simulated by the global Community Earth System Model-Last Millennium Ensemble (CESM-LME) (Yang et al., 2021), which demonstrate the spatial consistency in precipitation across the northeastern of the Tibetan Plateau, the Indian subcontinent, and eastern China (Figure S17 in their paper). We have included this study and the associated reference in the revised version of the manuscript. For historical times, climate reconstructions of warm-season precipitation over the last 531 years reveal that the dominant mode variability is a monopole covering most of China, including both the northeastern parts of the Tibetan Plateau as well as the eastern Tibetan Plateau region (Figure R7, Shi et al., 2017, 2020).

Please see Line 301–311, as well as Supplementary Fig. 7.

Figure R6. Spatial correlation maps of Tibetan Plateau precipitation reconstruction with the precipitation datasets from the CRU TS4.08 and GPCCC.

[Redacted]

Figure R7. Comparison of the proxy-based reconstructed (a1–b3) and instrumental precipitation fields for China (c1–c3) during the period May–September based on empirical orthogonal function (EOF) analyses (This figure was from Shi et al. 2019).

2) multi-decadal to centennial scales:

Our study primarily focuses on highlighting that the three phases of reduced precipitation/drier conditions occurred predominantly on multi-decadal to centennial scales. These long-term climate change trends are not adequately captured by modern observational data due to their limited data coverage back in time. Our multi-millennial TP precipitation reconstruction is consistent with existing precipitation reconstructions in both the East Asian monsoon region and westerly dominated region (Supplementary Fig. 6) on multi-decadal to millennial scales, and thereby demonstrates a significant and common hydroclimate consistency between western and eastern China.

Thus, the multi-decadal to millennial patterns in our long-term precipitation reconstruction for the northeastern Tibetan Plateau offer valuable insights into the historical climate conditions that influenced the development of civilizations in China.

References:

- Chen, F.H., et al., 2015. Agriculture facilitated permanent human occupation of the Tibetan Plateau after 3600 BP. *Science*, 347, 248–250.
- Shi, F., et al., 2017. Multi-proxy reconstructions of May–September precipitation field in China over the past 500 years. *Climate of the Past*, 13, 1919–1938.
- Shi, F., et al., 2019. Monopole mode of precipitation in East Asia modulated by the South China Sea over the last four centuries. *Geophysical Research Letters*, 46,

14713–14722.

Yang, B., et al., 2021. Long-term decrease in Asian monsoon rainfall and abrupt climate change events over the past 6,700 years. *Proceedings of the National Academy of Sciences*, 118, e2102007118.

Lines 118-122: The argument is problematic. It is true that tree-ring oxygen record can reflect large-scale atmospheric circulation patterns in tropical areas. However, it is questionable in this study region due to strong evaporative leaf water enrichment. As the authors wrote: Decreases in precipitation, typically associated with increasing aridity, lead to higher 18O enrichment in leaf and soil water due to elevated atmospheric vapor pressure deficit.

[Response]

To our knowledge, there is currently no evidence that suggests that only tree-ring oxygen isotopes in tropical regions could reflect large-scale atmospheric circulation. Studies in Europe utilized tree-ring $\delta^{18}\text{O}$ to reconstruct drought history and explore its relationship with large-scale circulation patterns (Andreu-Hayles et al., 2017; Nagavciuc et al., 2019). Tree-ring $\delta^{18}\text{O}$ from extra-tropics of south America reflected the Southern Annular Mode and the strength of the Westerlies (Grießinger et al., 2018). For semi-arid China, several studies also showed a connection between tree-ring $\delta^{18}\text{O}$ and large-scale atmospheric circulation systems, such as the Eastern Asian Summer Monsoon and the Westerlies. For example, Xu et al. (2019) found tree-ring $\delta^{18}\text{O}$ from arid central Asia reflected the drought variability and was reflected by the westerly circulation, shedding light on atmospheric circulation dynamics in central Asia. A nearby short tree-ring $\delta^{18}\text{O}$ chronology (approximately 100 km from our study site) indicating a significant relationship with the East Asian Summer Monsoon (Xu et al., 2011). This connection was also evident in $\delta^{18}\text{O}$ chronologies from arid to semi-arid regions on the western margin of the East Asian Summer Monsoon (Li et al., 2019, 2020).

According to McCarroll and Loader (2004) and Gessler et al. (2014), the oxygen isotope fractionation in tree rings reflects a combination of source water isotopes and local evaporation fractionation. The $\delta^{18}\text{O}$ values in tree-ring cellulose are influenced by the isotopic ratio of precipitation, which responds to atmospheric pressure patterns during the summer. Further fractionation occurs in the leaf, driven primarily by differences between internal and external water vapor pressures and, thus, by relative humidity. Dry summer conditions result in higher $\delta^{18}\text{O}$ values in leaf sugars and, consequently, in wood cellulose (Nagavciuc et al., 2022). Importantly, in regions with wet summers, tree-ring $\delta^{18}\text{O}$ is predominantly influenced by source water $\delta^{18}\text{O}$ (Treydte et al., 2014). On the northeastern Tibetan Plateau characterized by arid climate, precipitation directly influences tree-ring $\delta^{18}\text{O}$, tree-ring oxygen isotopes can effectively reflect regional atmospheric circulation patterns. Previous studies have established connections between tree-ring $\delta^{18}\text{O}$ on the Qinghai-Tibet Plateau and different large-scale atmospheric circulation patterns (Wernicke et al., 2017; Xu et al., 2011; Yang et al., 2021), further supporting this conclusion.

References:

- Andreu-Hayles, L., et al., 2017. 400 Years of summer hydroclimate from stable isotopes in Iberian trees. *Climate Dynamics*, 49, 143–161.
- Gessler, A., et al., 2014. Stable isotopes in tree rings: towards a mechanistic understanding of isotope fractionation and mixing processes from the leaves to the wood. *Tree Physiology*, 34, 796–818.
- Griessinger, J., et al., 2018. Imprints of climate signals in a 204 year $\delta^{18}\text{O}$ tree-ring record of *Nothofagus pumilio* from Perito Moreno Glacier, Southern Patagonia (50°S). *Frontiers in Earth Science*, 6, 27.
- Li, Q., et al., 2020. Oxygen stable isotopes of a network of shrubs and trees as high-resolution palaeoclimatic proxies in Northwestern China. *Agricultural and Forest Meteorology*, 285–286, 107929.
- Li, Q., et al., 2019. East Asian Summer Monsoon moisture sustains summer relative humidity in the southwestern Gobi Desert, China: evidence from $\delta^{18}\text{O}$ of tree rings. *Climate Dynamics*, 52, 6321–6337.
- McCarroll, D., Loader, NJ, 2004. Stable isotopes in tree rings. *Quaternary Science Reviews*, 23, 771–801.
- Nagavciuc, V., et al., 2019. Stable oxygen isotopes in Romanian oak tree rings record summer droughts and associated large-scale circulation patterns over Europe. *Climate Dynamics*, 52, 6557–6568.
- Nagavciuc, V., et al., 2022. A ~700 years perspective on the 21st century drying in the eastern part of Europe based on $\delta^{18}\text{O}$ in tree ring cellulose. *Communications Earth & Environment*, 3, 277.
- Treydte, K., et al., 2014. Seasonal transfer of oxygen isotopes from precipitation and soil to the tree ring: source water versus needle water enrichment. *New Phytologist*, 202, 772–783.
- Wernicke, J., et al., 2017. Multi-century humidity reconstructions from the southeastern Tibetan Plateau inferred from tree-ring $\delta^{18}\text{O}$. *Global and Planetary Change*, 149, 26–35.
- Xu, G.B., et al., 2011. Potential linkages between the moisture variability in the northeastern Qaidam Basin, China, since 1800 and the East Asian summer monsoon as reflected by tree ring $\delta^{18}\text{O}$. *Journal of Geophysical Research-Atmospheres*, 116, D09111.
- Xu, G.B., et al., 2019. Regional drought shifts (1710-2010) in East Central Asia and linkages with atmospheric circulation recorded in tree-ring $\delta^{18}\text{O}$. *Climate Dynamics*, 52, 713–727.
- Yang, B., et al., 2021. Long-term decrease in Asian monsoon rainfall and abrupt climate change events over the past 6,700 years. *Proceedings of the National Academy of Sciences*, 118, e2102007118.

Lines 154-157: the citations that shows carbohydrates formed in the previous year substantially contribute to earlywood formation are not necessarily applicable to the Tibetan Plateau.

[Response]

Thank you for the suggestion, we have deleted this part in the new revision.

Lines 200-203: the authors use tree-ring-width data from north-central China, stalagmite records from the southeastern Tibetan Plateau and lake sediment records from central Asia to verify the extreme drought intensity in the 20th century. Such comparisons are inadvisable because precipitation variations differ regionally.

[Response]

As previously discussed, precipitation exhibits variability on short-term scales, such as hours, days, and even annual cycles, across large regions. However, over longer periods such as multi-decadal and centennial scales, similarities in precipitation patterns become evident across extensive areas. Comparing climate data from different proxies over large spatial regions is a standard practice in paleoclimate studies and helps improve our understanding of climate change patterns and mechanisms. Numerous examples could be cited to support this. Four high quality papers are provided as following.

The $\delta^{18}\text{O}$ of carbonate from Lake Bosumtwi in West Africa was compared with the speleothem record from Southern China and the sea sediment record from the Arabian Sea (Shanahan et al., 2009). Comparisons were also made between the speleothem record from Wanxiang Cave in western China, the Longxi drought/flood index derived from historical documents in western China, and Alpine glacial records located in remote Europe (Zhang et al., 2008). Furthermore, tree-ring $\delta^{18}\text{O}$ records from northern Pakistan were compared with tree-ring-based annual precipitation reconstructions in northeast China, southwest Asian monsoon intensity inferred from *Globigerina* bulloids in the Arabian Sea, drought reconstructions based on tree rings in western USA, as well as spring-summer precipitation reconstructions derived from tree-rings in southern Germany (Treydte et al., 2006). Tree-ring $\delta^{18}\text{O}$ records from Japan was compared with Northern Hemisphere temperature records, two speleothem records from China, as well as precipitation reconstructions based on the sediment cores from Tsuifong Lake in Taiwan (Figure 17 in Nakatsuka et al., 2020).

These examples highlight the value of using diverse climate proxies to gain insights into historical climate variability and patterns.

References:

- Nakatsuka, T., et al., 2020. A 2600-year summer climate reconstruction in central Japan by integrating tree-ring stable oxygen and hydrogen isotopes. *Climate of the Past*, 16, 2153–2172.
- Shanahan, T.M., et al., 2009. Atlantic forcing of persistent drought in West Africa. *Science*, 324, 377–380.
- Treydte, K.S., et al., 2006. The twentieth century was the wettest period in northern Pakistan over the past millennium. *Nature*, 440, 1179–1182.
- Zhang, P.Z., et al., 2008. A test of climate, sun, and culture relationships from an 1810-year Chinese cave record. *Science*, 322, 940–942.

Lines 208-223: the authors used the simulation data from Community Earth System Model Last Millennium Ensemble (CESM-LME) project to demonstrate the 20th century decreasing trend in precipitation. One wonders if the model performs well in this high-elevation area of unusually complex terrain.

[Response]

The CESM-LME has been found capable to realistically simulate multiple scale responses of temperature and Asian summer monsoon precipitation over the Tibetan Plateau to external forcings during the last millennium (Chen et al., 2024; Liu et al., 2021; Zuo et al., 2024). Furthermore, multi-model ensemble mean precipitation changes over northern Tibetan Plateau from the Aerosol Chemistry Model Intercomparison Project (DAMIP) from the Coupled Model Intercomparison Project Phase 6 (CMIP6) (Collins et al., 2017) also confirm that this precipitation decline is induced by anthropogenic aerosol emissions. These findings have now been incorporated into our main text in the revised version (Figure R8), please see line 228–231 and Supplementary Fig. 3.

Figure R8. Precipitation simulations of different climate models over the Tibetan Plateau (34–39°N, 94–102°E). **a**, Tibetan Plateau precipitation in CESM-LME full-forcing simulation; **b**, Tibetan Plateau precipitation in only Ozone-aerosol forcing simulation; **c**, Tibetan Plateau precipitation in CMIP6 multi-model aerosol-only forcing simulation. Blue line represents the mean, shadow area represents mean \pm 1 standard deviation, red line represents the interannual trend, respectively.

References:

Chen, P.J., et al., 2024. Impact of solar activity and ENSO on the early summer Asian Monsoon during the last millennium. *Geophysical Research Letters*, 51,

e2023GL105668.

Collins, W.J., et al., 2017. AerChemMIP: quantifying the effects of chemistry and aerosols in CMIP6. *Geoscientific Model Development*, 10, 585–607.

Liu, W., et al., 2021. Impact of volcanic eruptions on hydro-thermal combination of the Tibetan Plateau and Arctic during mid-fifteenth century. *Quaternary Sciences*, 41, 714–725. (in Chinese)

Zuo, M., et al., 2024. Understanding surface temperature changes over the Tibetan Plateau in the last millennium from a modeling perspective. *Climate Dynamics*, 62, 5483–5499.

In addition, it is strange how the authors concluded that the combined weakening of both the Asian Monsoon and Westerlies has led to this severe aridity in China during the 20th century. It is also inappropriate to overemphasize human activity (i.e., increased anthropogenic aerosol emissions) impact on the climate system.

[Response]

Firstly, previous studies have demonstrated that the precipitation over the Tibetan Plateau is influenced by both, the Asian Monsoons and the Westerlies (An et al., 2019; Chen et al., 2016; Cui et al., 2021; Massimo et al., 2011; Yu et al., 2016). In our study, we found that the reconstructed TP precipitation exhibits consistency with hydroclimate variability from the Westerlies (Lan et al., 2019) and Asian monsoon regions (Cai et al., 2017; Liu et al., 2019; Tan et al., 2018), in particular on multi-decadal to centennial scales (Supplementary Fig. 6). These findings support the idea that the precipitation on the Tibetan Plateau is influenced by the westerly and monsoon systems. Our precipitation series additionally reveals a significant decline towards drier conditions during the past century, a trend which is also observed in many precipitation records from the Westerlies and Asian monsoon regions (Supplementary Figs. S5 and S6). Therefore, we attributed the weakening of both the Asian Monsoon and Westerly systems to a large-scale aridity trend during the 20th century.

Regarding the impact of anthropogenic aerosol emissions, numerous studies have demonstrated their significant effects on the Asian monsoons (Bollasina et al., 2011; Liu et al., 2019) and the westerlies in the Northern Hemisphere (Dong et al., 2022; Williams et al., 2020; Wu et al., 2013; Zhang et al., 2024), with resulting global implications (Chiang et al., 2021). These top-tier journal publications should not be ignored. Our simulation results further corroborate these findings. While it is important to approach the attribution of climate change to human activities with respective scientific objectivity and caution, our results are consistent with other research supporting the impact of anthropogenic aerosols on global and regional climate.

References:

An, Z.S., et al., 2019. Severe haze in northern China: A synergy of anthropogenic emissions and atmospheric processes. *Proceedings of the National Academy of Sciences*, 116, 8657–8666.

- Bollasina, M.A., et al., 2011. Anthropogenic aerosols and the weakening of the South Asian summer monsoon. *Science*, 334, 502–505.
- Cai, W.J., et al., 2017. Weather conditions conducive to Beijing severe haze more frequent under climate change. *Nature Climate Change*, 7, 257–262.
- Chen, F.H., et al., 2016. Holocene moisture and East Asian summer monsoon evolution in the northeastern Tibetan Plateau recorded by Lake Qinghai and its environs: A review of conflicting proxies. *Quaternary Science Reviews*, 154, 111–129.
- Chiang, F., et al., 2021. Evidence of anthropogenic impacts on global drought frequency, duration, and intensity. *Nature Communications*, 12, 2754.
- Cui, A.N., et al., 2021. Tibetan Plateau precipitation modulated by the periodically coupled westerlies and Asian Monsoon. *Geophysical Research Letters*, 48, e2020GL091543.
- Dong, B.W., et al., 2022. Recent decadal weakening of the summer Eurasian westerly jet attributable to anthropogenic aerosol emissions. *Nature Communications*, 13, 1148.
- Lan, J.H., et al., 2019. Late Holocene hydroclimatic variations and possible forcing mechanisms over the eastern Central Asia. *Science China-Earth Sciences*, 62, 1288–1301.
- Liu, Y., et al., 2019. Anthropogenic aerosols cause recent pronounced weakening of Asian Summer Monsoon relative to last four centuries. *Geophysical Research Letters*, 46, 5469–5479.
- Massimo, M.A., et al., 2011. Anthropogenic aerosols and the weakening of the South Asian Summer Monsoon. *Science*, 334, 502–505.
- Tan, L.C., et al., 2018. High resolution monsoon precipitation changes on southeastern Tibetan Plateau over the past 2300 years. *Quaternary Science Reviews*, 195, 122–132.
- Williams, A.P., et al., 2020. Large contribution from anthropogenic warming to an emerging north American megadrought. *Science*, 368, 314–318.
- Wu, P.L., et al., 2013. Anthropogenic impact on Earth's hydrological cycle. *Nature Climate Change*, 3, 807–810.
- Yu, S.C., et al., 2016. Anthropogenic aerosols are a potential cause for migration of the summer monsoon rain belt in China. *Proceedings of the National Academy of Sciences*, 113, E2209–E2210.
- Zhang, W.X., et al., 2024. Anthropogenic amplification variability over the past century. *Science*, 385, 427–432.

Lines 234-235: the authors used the word 'local'. Actually, the Monsoon Asia Drought Atlas 235 (MADA) is not local but regional or at a much larger spatial scale.

[Response]

Thank you for this valuable comment. In fact, we conducted a comparison using the nearest grid point from the MADA atlas located close to our study region. In order to ensure accuracy and prevent any potential misinterpretation, we have modified the sentence as follows: “On an annual scale, our precipitation reconstruction exhibits a strong correlation with the local drought reconstruction derived from the Monsoon Asia

Drought Atlas41 (MADA) within our study region”

Please see Line 257–259.

Lines 234-252: these comparisons mean not much to me since the correlation is so weak with maximum explained variance being 0.2-0.3. Moreover, it should be noted that these low-resolution proxy records such as stalagmites, lake sediments, and ice cores have large dating uncertainties. Correlation does not mean causation, especially when only a small fraction of the available data is shown.

[Response]

“Correlation” and “variance” are distinct statistical measures, each serving different purposes. In paleoclimatology and modern meteorology, the consistency between two series is typically assessed based on the correlation coefficient rather than the magnitude of explained variance. The correlation coefficient, which reflects the degree of association between two datasets, is influenced by sample size. Even with a low correlation coefficient, a large sample size can yield statistically significant results, demonstrating a meaningful relationship between the sequences.

Our tree-ring $\delta^{18}\text{O}$ series shows significant correlations with drought reconstructions based on tree-ring width from our study region and with other nearby $\delta^{18}\text{O}$ series. While these comparisons involve relatively short records, they cannot match the length of our long-term dataset. In this context, stalagmite and lake records, although lower in resolution, provide valuable low-frequency climate signals compared to tree-ring width chronologies. Thus, comparing these records with our Tibetan Plateau tree-ring $\delta^{18}\text{O}$ series is appropriate for examining hydroclimate variability on multi-decadal to centennial scales and a cross-proxy comparison is state of the art in modern paleoclimatology.

Given the influence of both the East Asian Summer Monsoon and the Westerlies on our study region, we compared our Tibetan Plateau precipitation reconstruction with hydroclimate series from these two dominant climate systems. The significant correlations observed indicate synchronous hydroclimate patterns over a broad spatial scale. These associations are likely not coincidental, suggesting a robust relationship between our reconstructed precipitation records and various hydroclimate records. This finding indicates consistent hydroclimate changes in areas influenced by both westerlies and the East Asian Summer Monsoon over the past 3,500 years.

I am concerned HOW the fraction of globally distributed records were chosen considering a large number of proxy records are available. As the authors wrote: These variations might also be due to the lower resolution and dating uncertainties associated with other archives. These uncertainties often stem from changes in carbon reservoir age in lacustrine sediments over time.

[Response]

In our study, we focused on comparing hydroclimatic records from regions dominated by the summer monsoon systems and the westerlies in East Asia, rather than focusing on a global approach. We selected regional records that exhibit distinct hydroclimatic signals and additionally offer high resolution and regional

representativity.

There is limited availability of hydroclimate records from the late Holocene, and many of these proxy face dating issues, with tree-ring based proxy records being a notable exception. This highlights the importance of reconstructing hydroclimate variability with a) accurate dating and b) high temporal resolution. To address this, we interpolated low-resolution stalagmite and lake data for comparison with our high-resolution tree-ring $\delta^{18}\text{O}$ series. This approach underscores the value of our 3,500-year TP record as a benchmark for evaluating other proxies.

Lines 282-288: It is hard to understand WHY and HOW the wet period identified in precipitation reconstruction from the 720s BC to the AD 70s could be linked to significant cultural developments-intellectual and philosophical advancements in ancient China. Correspondence does not imply causation. The same problem is with the lines 289-298, 299-312 and 313-318. When linking the rise and fall of dynasties to climate system, one must be aware of the spatial coherence of climate change.

[Response]

The relationship between climate and civilization has been subject of extensive scholarly research (see the references below). It is unclear and strange to us why the reviewer questions this established link.

Climate change researches are crucial due to its profound impact on societal stability and development. Such studies aim to reconstruct historical climate patterns, understand the mechanisms driving climate variations, forecast future trends, and ultimately support sustainable development.

Rhoads Murphey's book "*A History of Asia*" begins with an introduction of the Asian monsoon, arguing that agriculture underpins social stability and that monsoon patterns shape agricultural practices in Asia. In low-productivity agricultural societies, where rain-fed crops depend directly on precipitation, any fluctuations in rainfall can significantly affect social stability and the progress of civilization. Numerous studies illustrate this relationship, such as the 7th-century BC collapse of Assyria (Ashish et al., 2019), the Hittite collapse around 1198–1196 BC (Manning et al., 2023), the decline of the Maya civilization (AD 800–1000) and the Ming Dynasty in the 1640s (Degroot et al., 2021), all of which correlate with severe megadroughts. The precipitation reconstruction based on low-resolution tree-ring $\delta^{18}\text{O}$ data in northeastern Tibetan Plateau indicates that periods of wet climate were associated with flourishing cultures, while dry conditions often coincided with the disintegration of historical civilizations (Yang et al., 2021).

Regarding to your comment "Correspondence does not imply causation", our study does not arbitrarily establish the connection between climate and historical events. In fact, the humid period from the 720s BC to the AD 70s and several drought periods noted in our manuscript are well corroborated by historical Chinese literatures. Climate reconstruction is a true and valuable tool of cross-validating historical records, historical prosperity, or declining in societies/cultures. Our quantitative precipitation reconstruction offers a detailed assessment of the interplay between past climate and corresponding agro-cultural changes. Favorable climate conditions with ample

precipitation support agricultural productivity, which in turn promotes social stability and prosperity, creating a vibrant cycle that fosters cultural development and civilizational progress (Fang et al., 2004). During periods of abundance, individuals are more likely to engage in philosophical and scientific inquiry, leading to cultural and technological advancements. Conversely, extreme droughts and other adverse climatic conditions can precipitate agricultural failures, food shortages, famines, and social unrest, ultimately hindering cultural and technological progress (Ge et al., 2011).

While climate change has significantly influenced the evolution of Chinese civilizations and the rise and fall of dynasties, it is essential to acknowledge the interplay of other factors such as economics, politics, agriculture, and military considerations. These elements also play crucial roles in shaping socio-political outcomes (Ge et al., 2011).

References:

- Carleton, T.A., et al., 2016. Social and economic impacts of climate. *Science*, 353, aad9837.
- Degroot, D., et al., 2021. Towards a rigorous understanding of societal responses to climate change. *Nature*, 591, 539–550.
- Dunlap & Brulle (Eds.), 2015. *Climate change and society: Sociological perspectives*. Oxford University Press.
- Fagan, B., 2019. *The Little Ice Age: how climate made history 1300-1850*. Hachette, UK.
- Fang, X.Q., et al. 2004. Progress and prospect of researches on impacts of environmental changes on Chinese civilization. *Journal of Palaeogeography*, 6, 85–94.
- Fleming, 2009. The Great Warming: Climate Change and the Rise and Fall of Civilizations. *Physics Today*, 62, 52–53.
- Ge, Q.S., et al. 2011. China's history of climate change. *Science Press*.
- Hsu, K.J. 1998. Sun, climate, hunger, and mass migration. *Science in China Series D-Earth Sciences*, 41, 449–472.
- Huntington, E. 1924. *Civilization and Climate*. Yale University Press.
- John, D., et al., 2014. *The Impact of Climate on Human Histories*. Jincheng Press.
- Lee, L. 2008. *Blame it on the Rain: How the Weather Has Changed History*. Harper Collins.
- Ma, D. 2017. *Climate Changes History*. Shanxi People's Publishing House.
- Manning, S.W., et al., 2023. Severe multi-year drought coincident with Hittite collapse around 1198-1196 BC. *Nature*, 614, 719–724.
- Murphey, R. 1950. *A History of Asia (7th ed.)*. Routledge.
- Sinha, A., et al., 2019. Role of climate in the rise and fall of the Neo-Assyrian Empire. *Science Advances*, 5, eaax6656.
- Yang, B., et al., 2021. Long-term decrease in Asian monsoon rainfall and abrupt climate change events over the past 6,700 years. *Proceedings of the National Academy of Sciences*, 118, e2102007118.

Reviewer #2 (Remarks to the Author)

The hydroclimate of the Tibetan Plateau is strongly influenced by both the presence of the Asian Monsoon and Westerlies. In recent years, there have been several pioneering papers reconstructing the climate of the region with the primary focus upon Asian monsoonal precipitation. Many of the co-authors of this paper are linked to this independent research. For example, Yang et al. (2021; not cited) published a 6,700-year isotope chronology from Delingha that represented the “longest existing precisely dated isotope chronology in Asia” typically with a one-to-five-year resolution. The novelty of THIS STUDY is that it presents the longest precipitation reconstruction from a 3476-year precisely dated annually resolved oxygen isotope chronology. The reconstruction allows current droughts to be investigated against natural climate variability over three millennia. It’s a word that gets used to often these days, but these droughts appear to be “unprecedented.” However, the time-series end in the year AD 2010 illustrating the need to continual chronology development.

[Response]

Thanks for your comments. In the future studies, we will update and extend the chronology.

We cited Yang et al. 2021’ paper in the new revision.

Reference:

Yang, B., et al. 2021. Long-term decrease in Asian monsoon rainfall and abrupt climate change events over the past 6,700 years. *Proceedings of the National Academy of Sciences*, 118, e2102007118.

The links to societal changes are speculative in places but cannot be discounted. Interpretation should be cautious especially when sample replication is low. Overall, the paper makes a significant contribution towards the field. I recommend publication subject to minor changes.

[Response]

Thanks for your positive comments.

L78. As the time-series end in AD2010, the authors should refer to 20th (and 21st century!) droughts.

[Response]

Thank you for your suggestion.

The 20th century drought is assessed from a centennial perspective, while it should be noted that precipitation has exhibited an increasing trend over the past decade.

L83-84. This comment appeared non-sequitur, but it is partly justified later in the text (L325-335). It is acceptable.

[Response]

Thank you. We have deleted this sentence in the new revision.

L135-144. Although it's not totally prescriptive, what was the typical EPS value found in THIS STUDY to constitute a robust dataset? This is important. It is also not covered in the Methods (L.473-478) and too difficult to read on Figure 2 (L602).

[Response]

The mean EPS value was 0.90, which was considerably larger than the recommended threshold of 0.85. In addition, we added a dashed line in Fig. 2 in the revised maintext to show the threshold of 0.85. For most periods, $ESP > 0.85$. We have added this in the Methods section. Please see line 397–398.

L144 (and later). It should be emphasised that any interpretation covering the first six hundred years of the time-series should be cautious as it is based upon a single tree. This is not uncommon but it should be acknowledged.

[Response]

We added a sentence to acknowledge that caution should be exercised when interpreting precipitation data for the first six hundred years (1465–849 BC) of the chronology, as it is based on only one tree. However, this period can still serve as a reference in the absence of other available data. Please see Line 155–158.

L193 (and later). This period also includes ten-years of the 21st century.

[Response]

As we mentioned above, the 20th century drought is assessed from a centennial perspective.

L462. Please provided supporting evidence for the homogenisation process. I presume that it is based upon Laumer et al. (2009).

[Response]

Yes, you are right. We provided the evidence for the homogenisation process and cited the reference of Laumer et al. (2009).

Reference:

Laumer, W., et al. 2009. A novel approach for the homogenization of cellulose to use micro-amounts for stable isotope analyses. *Rapid Communications in Mass Spectrometry*, 23, 1934–1940.

L499-502. What is the possible cause of the observed relationship?

[Response]

We added explanations in the revision:

The tree-ring $\delta^{18}\text{O}$ is mainly dependent on source water $\delta^{18}\text{O}$ and local evaporation fractionation (McCarroll and Loader, 2004). In our study region characterized by arid climate, the source water is predominantly determined by precipitation. Decreases in precipitation, typically associated with increasing aridity, may increase evapotranspiration and consequently lead to higher ^{18}O enrichment in leaf and soil water due to elevated atmospheric vapor pressure deficit. Please see Line 166–173.

Reference:

McCarroll, D., Loader, N.J., 2004. Stable isotopes in tree rings. *Quaternary Science Reviews*, 23, 771–801.

L515 (& EXTENDED L156). The periods used for the split calibration-verification are not truly independent (AD 1961–1990 and AD 1981–2010). Is this approach acceptable?

[Response]

Yes, this approach based on previous studies is acceptable. In statistical analysis, at least 30 years are required to establish the relationship between two variables (Wei, 2007). Considering the relatively limited duration of available meteorological data for our study region spanning only 50 years, we conducted a split test employing a calibration period of 30 years and a validation period of 20 years. Specifically, within this study, we utilized AD 1961–1990 as the calibration period and AD 1991–2010 as the verification period; similarly, we employed AD 1981–2010 as the calibration period and AD 1961–1980 as the verification period. This methodology has been already successfully validated (Fang et al., 2010).

Reference:

Fang, K.Y., et al., 2010. Tree-ring based drought reconstruction for the Guiqing Mountain (China): linkages to the Indian and Pacific Oceans. *International Journal of Climatology*, 30, 1137–1145.

Wei, F.Y., 2007. Statistical Diagnosis and Prediction Technique Applied in Modern Climatology. *China Meteorological Press, Beijing*.

L590. Figure 1a. Carefully re-select the colors as yellow font is difficult to read on a colored background.

[Response]

Thank you for this valuable comment. We changed the colors according to your suggestion.

L602. Figure 2. Strictly there is no zero in the AD/BC timescale as AD refers to in the “year of Our Lord” so 1BC precedes AD1. Many scientists now use the on-secular Common Era timescale (BCE/CE) but the same issue remains. The approach adopted here has been used elsewhere, but how do the authors justify the use of zero as it doesn’t exist.

[Response]

Yes, the utilization of the concept of "year zero" was employed in our study. The incorporation of this notion holds significant importance within interdisciplinary investigations pertaining to climate and historical analyses (Büntgen et al., 2020, PNAS). In accordance with this chronology, the year preceding AD 1 was designated as "year zero," while the second year prior to AD 1 was denoted as 1 BC.

Reference:

Büntgen, U., et al., 2020. The importance of "year zero" in interdisciplinary studies of climate and history. *Proceedings of the National Academy of Sciences*, 117, 32845–32847.

L602. Clarity required with the EPS and Rbar values. At how many points is EPS less than the generally accepted value of 0.85?

[Response]

The dashed line in Figure 2b has been incorporated to indicate the EPS threshold of 0.85. Four periods, centered at 435 BC, AD 715 (0.76), AD 815 (0.82), and AD 1715 (0.84), exhibit values below this threshold. We have added this in the Methods section. Please see line 399–401.

EXTENDED L135. Sample XTT is from a lower elevation. Will it still respond to the same drivers?

[Response]

The sample obtained from XTT is an archaeological wood sample originating from a low-altitude ancient tomb. However, considering that current juniper forests in this region are only found at elevations ranging between approximately 3500–4100m (Shao et al., 2010), this wood would have presumably been collected from higher sites. Consequently, it can be inferred that the elevation characteristics of archaeological wood align with those of both living and dead trees in SG site.

Please see line 149–154, 385–390 in the maintext, and the supplementary note.

Reference:

Shao, X. et al., 2010. Climatic implications of a 3585-year tree-ring width chronology from the northeastern Qinghai-Tibetan Plateau. *Quaternary Science Reviews*, 29, 2111–2122.

EXTENDED L155. The period covered should be AD 1961-2010.

[Response]

It was changed.

Recommended additional references:

Laumer, W., et al. (2009). "A novel approach for the homogenization of cellulose to use micro-amounts for stable isotope analyses." *Rapid Communications in Mass Spectrometry* 23(13): 1934-1940.

Yang, B., et al. (2021). "Long-term decrease in Asian monsoon rainfall and abrupt climate change events over the past 6,700 years." *Proceedings of the National Academy of Sciences* 118(30): e2102007118.

[Response]

Thank you for this suggestion. The references have been added in the new version of our manuscript.

REVIEWER COMMENTS

Reviewer #1 (Remarks to the Author):

I appreciate Authors' effort to improve the original submission during the revision stage. However, I do see some flaws that would prohibit publication in its current form.

One of the most important conclusions of this manuscript is that recent drought on the Tibetan Plateau is unprecedented in the past 3500 years. It's crucial to ensure that such statements are supported by robust evidence. The title should reflect the findings accurately and not overstate them based on the current data. What is meant by the word 'recent'?

[Response]

We revised the title to "Recent centennial drought on the Tibetan Plateau is unprecedented in the past 3500 years". The reasons are as following:

Since the earliest ca. 600-yr $\delta^{18}\text{O}$ segment is based on only one individual tree, we conducted a comparative analysis with the existing Yang's precipitation reconstruction, which can be regarded as a regional benchmark. This validation supports the reliability of our findings, unless there are serious reservations about the reliability of Yang's data. Our analysis revealed that the average precipitation during AD 1930–2010 was the lowest observed in the past 3485 years on a centennial scale (Line 223 in the main text). Therefore, in the context of 3500 years, "Recent" refers to approximately one century, as commonly employed in high-quality dendroclimatological literature. However, we have included "centennial" in the title to clearly indicate the time frame of the "unprecedented" recent drought.

Examples from the references:

Büntgen, U., et al. (2021). "Recent European drought extremes beyond Common Era background variability." *Nature Geoscience* 14(4): 190-196.

Liu, Y., et al. (2017). "Recent enhancement of central Pacific El Nino variability relative to last eight centuries." *Nat Communications* 8: 15386.

Trouet, V., et al. (2018). "Recent enhanced high-summer North Atlantic Jet variability emerges from three-century context." *Nature Communications* 9.

In general, the oxygen isotopic ratio varies greatly with elevation, i.e., altitude effect. I would suggest the authors exemplifies how the isotope mean of each sample varies over time and with elevation. In addition, an altitude-isotope relationship should be plotted to visually inspect if the altitude effect exists or not.

[Response]

Thank you for this valuable and important comment. Previous investigations consistently demonstrated no altitude-related effects on tree-ring $\delta^{18}\text{O}$ values in Qilian junipers on the northern Tibetan Plateau (Xu et al., 2017; Yang et al., 2021), as mentioned in our previous revision. Mostly notably, Yang et al. (2021) clearly pointed out that "no significant differences were detected in the mean or SD of tree-ring $\delta^{18}\text{O}$

values at different elevations” within the same study area as ours. Since both studies originate from a geographically closely situated area, and since both isotope series are very similar, we can expect the same relationships for our series.

However, to carefully take into account the reviewer's concern and suggestion, we conducted the requested comparison shown in the figure below. It is important to emphasize that long-term climate changes can cause variations in absolute $\delta^{18}\text{O}$ values in trees, making it inappropriate and in our sense meaningless to compare absolute tree-ring $\delta^{18}\text{O}$ values across different time spans. Therefore, our analysis of the altitude-isotope relationship was limited to only seven trees within the most recent common period from AD 1835-1974, further demonstrating the absence of altitude-related effects on tree-ring $\delta^{18}\text{O}$ values in our study region.

Figure R1 Visualization of the relationship between average tree-ring $\delta^{18}\text{O}$ values from this study with altitude. a, mean $\delta^{18}\text{O}$ value and lengths of each individual tree. b, elevations and lengths of each individual tree (blue lines indicating samples sharing the same elevation as other one), except for the archeological woods due to unknown true elevation. c, altitude-isotope relationship among the seven trees within the recent common period AD 1835-1974.

The "outliers correction method" doesn't seem to answer my concerns. The authors stated that they found no outlier trees in their study, suggesting no significant offset among the 17 $\delta^{18}\text{O}$ tree-ring series that were analyzed. That being the case, how can you come to that conclusion “Recent drought on the Tibetan Plateau is unprecedented in the past 3500 years”?

[Response]

We think that our statement was misinterpreted, leading Reviewer 1 to draw an erroneous conclusion of “no significant offset among the 17 $\delta^{18}\text{O}$ tree-ring series that were analyzed”.

The "Outliers correction method" was introduced by Arosio et al. (2024) and employed to identify outlier trees that exhibit significant divergence from the majority of observations within their respective timespans.

Out-of-range offset trees were identified as those with mean values exceeding or falling below the rolling mean of the same period plus or minus one standard deviation of the calibration period. The detection of each tree was conducted based on a comparative analysis within a **common time frame** shared among other trees, rather than comparing

every single tree individually. The tree-ring $\delta^{18}\text{O}$ values exhibit temporal variations, rendering the comparison of tree-ring $\delta^{18}\text{O}$ values across different periods inconsequential.

The expression “recent drought” is based on our robust precipitation reconstruction accounting for the long-term variability of the tree-ring $\delta^{18}\text{O}$. The superiority of tree-ring $\delta^{18}\text{O}$ over tree-ring width parameter is embodied in the long-term trend $\delta^{18}\text{O}$ exhibited. The tree-ring width chronology is usually quite flat due to the pre-analysis application of various detrending methods that always remove the typical long-term trend of climate. This does not apply to $\delta^{18}\text{O}$ series, which retain all low-frequency components found in the original data.

Reference:

Arosio, T. et al. Methodological constraints of tree-ring stable isotope chronologies. *Quat. Sci. Rev.* 340, 108861 (2024).

The provenance and growing elevation of the XTT and DGM archaeological wood, covering the period from 1465 BC to 86 BC, are important aspects that need clarification. From Supplementary Fig. 8, the overall $\delta^{18}\text{O}$ series encompassing all samples are notably positively biased than the tree-ring $\delta^{18}\text{O}$ chronology at the SG site. It will inevitably lead to biases and non-homogeneity of the final chronology. I agree with the second reviewer on the comment ‘any interpretation covering the first six hundred years of the time-series should be cautious as it is based upon a single tree’. It can barely be used to past-present comparison. Taking all these points into consideration, the credible time period for the chronology seems to be limited to the last 2000 years.

[Response]

Thank you very much for carefully pointing out this concern.

In respect to the concerns about archaeological wood, we want to help to overcome the expressed concerns: Archaeological wood was sourced from ancient tombs at lower elevations where no forests were present. The juniper forests only grow on slopes with elevations between 3500–4100 meters (Shao et al., 2010). Consequently, it can be inferred that the archaeological wood originated from higher elevations similar to those of SG sites. Despite the archaeological wood from different sites, they overlap the time span of dead trees from SG. The application of the 'outliers correction method' revealed no significant deviation between the archaeological wood and dead trees during the overlapping period. It is very difficult to discern a difference between the overall $\delta^{18}\text{O}$ series and the SG-derived $\delta^{18}\text{O}$ series in simply viewing of Fig. S8.

In case of any uncertainties regarding the archaeological wood, it should be noted that the SG chronology covers a time span from 761 BC to AD 2010, which is contrary to the reviewer statement that “the credible time period for the chronology seems to be limited to the last 2000 years”.

Additionally, we compared our precipitation reconstructions with the time series established by Yang et al. (2021) for the period between 1466 and 500 BC, clearly revealing similar variations between both series (Figure R2 a). In our understanding,

this further corroborate the reliability of our precipitation reconstruction during this period despite the sample replication of only one individual tree. However, we still included this sentence in the main text “However, caution should still be exercised when interpreting our precipitation data for the period during 1466–850 BC due to the limited replication.” (Line 189-191 in the main text).

To sum up, evidence shows our entire 3500-yr tree-ring $\delta^{18}\text{O}$ series is reliable, particularly with validation from Yang’s series.

Figure R2 Comparison of tree-ring $\delta^{18}\text{O}$ -based precipitation reconstructions between this study (blue) and the one developed by Yang et al. (2021) (orange) for the periods 1466–500 BC (a), AD 1800–2010 (b), and 1466 BC–AD 2010 (c), respectively.

The second reviewer mentioned that Yang et al. (2021) published a 6,700-year isotope chronology from Delingha that represented the “longest existing precisely dated isotope chronology in Asia” typically with a one-to-five-year resolution. Considering the dataset is from the same study area, it is essential and beneficial to add a figure showing how different the two chronologies are.

[Response]

Thank you in pointing out this issue. In general, the two sequences exhibit some similarities in terms of low-frequency variation, which further validates the reliability of our data, particularly the earliest 600 years derived from a single tree. Unfortunately,

due to the majority of Yang's data only having a temporal resolution of 3-5 years, it is not possible to make a sound comparison on an annual scale between the two series.

1) The most significant difference between the two series lies in their temporal resolution.

Our dataset provides year-by-year annual resolution, whereas Yang's data is characterized by variable resolution. According to data downloaded from the NOAA website, their sequence maintains an annual resolution at only 1951-2011, with a 3-year resolution during AD 0-1950 and a 5-year resolution before year zero. In contrast, our analysis encompasses a total of 15,028 individual tree-ring $\delta^{18}\text{O}$ values spanning 3,476 years, while Yang analyzed 9,526 $\delta^{18}\text{O}$ values spanning over a period of 6,700 years.

Thus, our sample size significantly surpasses that of Yang's. The tree-ring proxy differs from other proxies primarily in terms of its annual resolution and accurate dating. This lower-resolution (3-5 year) makes it more comparable to the stalagmite proxy, although inferior at times, but lacks certain advantages associated with tree-ring proxy.

It is more difficult to use Yang's data to assess detailed climate change information such as evaluating severity and frequency of extreme climate events, because mixing three or five-year tree-ring samples together can weaken the precise recording of annually appearing climate phenomena like multi-monthly drought / humid periods by averaging extreme years.

Additionally, the calculation for determining frequencies of extreme drought years is also impossible. As a result, their conclusions do not align closely with China's climate history. For example, they fail to account for the severe drought that occurred during the Xin Dynasty (AD 9–23), accompanied by the conspicuous appearance of cannibalism, which led to the downfall of regime. Another example is the severe multi-decadal drought period during the Three Kingdoms period (AD 220–280), that caused a catastrophic and rapid drop in China population from 60 million to 30 million.

Furthermore, the approaches employed to construct the final composite tree-ring $\delta^{18}\text{O}$ chronology differ significantly between the two studies.

Our final isotope chronology was generated by annually resolved individual tree-ring $\delta^{18}\text{O}$ series, whereas Yang's series exhibits inhomogeneities in its composition. Specifically, for the period between 1168-2011 in Yang's data, tree-ring samples were pooled, while individual tree-ring samples were utilized for other periods. The flaws caused by the pooling method are as following:

- 1) During AD 1168-2011, only one data point is available for every three years, making error detection and uncertainty calculation for the precipitation reconstruction impossible.
- 2) A considerably larger variability is observed during the period based on pooled samples compared to other periods.

Analyzing pooled sequences may obscure series characteristics due to sample mixing. Studies have shown that pooling processes can introduce bias due to uneven sample sizes and incomplete homogenization, potentially impacting result validity. Moreover, conducting only one measurement per year without replication makes it impossible to detect errors in the pooling method. In summary, sequential analysis of individual rings

is offers many advantages in tree-ring $\delta^{18}\text{O}$ studies.

A particularly clear distinction between our record and the Yang et al. lies in the fact that the driest periods in Yang's data occurred during AD 346-760, whereas ours took place in the 20th century.

This discrepancy is likely attributable to significant heterogeneity present in Yang's data, as discussed earlier, including the employment of a pooled method and the differing resolution of 1-3-5 years. When comparing the pooled series to individual chronologies in Yang's data (Figure R3), discrepancies emerge regarding the low-frequency trends. Notably, while the pooled sequence appears relatively flat, the individual sequences exhibit a more pronounced downward trend - similar to our own sequence. This elucidates why our time series demonstrates a declining trend while Yang's displays an increasing disparity over the past century.

Undoubtedly, the multi-millennial climate reconstruction by Yang et al. offers a valuable high-resolution window into hydroclimate variations since the mid-Holocene. Moreover, Yang's record is one of several we can use to validate the reliability of our reconstruction. Nevertheless, due to the reasons stated above, we feel that our consistently annually resolved isotope chronology presents a progress compared to Yang's time series.

[Redacted]

Figure R3 Comparison of $\delta^{18}\text{O}$ measurements obtained via averaging isotope series derived from individual trees (black line) and the pooling (red line) methods (This figure is copied from the supplementary information of Yang et al. 2021).

The authors used CESM-LME and CMIP6 multi-model aerosol-only forcing simulation

to demonstrate the 20th century decreasing trend in precipitation. However, it is well accepted that the climate was warm-dry and then changed to warm and wet from the end of the 19th century to the 1970s. Obviously, the model performance is flawed in simulating precipitation variations.

[Response]

First of all, it is unfortunately not clear to which region, season and study the reviewer is referring here, However, it seems that the statement “However, it is well accepted that the climate was warm-dry and then changed to warm and wet from the end of the 19th century to the 1970s” is derived from the abstract of Shi et al. (2007). Actually, they also stated that “From the end of the Little Ice Age to about 1980, the climate in northwest China has been warm-dry.”, which also demonstrate our initiated discussion about the recent centennial drought on the northeastern Tibetan Plateau.

Actually, the aerosol-only forcing simulation mainly serves as a sensitivity test aimed at analyzing the impact and mechanism of single aerosol forcing, rather than simulating actual climate changes. Our simulations demonstrate a decline in precipitation since AD 1850, with the trend being particularly pronounced under aerosol-only forcing. The humid period constitutes less than one-fifth of our entire simulation. Although recent precipitation has been increasing during the past two decades, the 20th century still remains the driest period at centennial scale. However, this recent increase in precipitation can only be regarded as minor fluctuations when compared to longer time scales.

Furthermore, many recent studies demonstrated a recent 20th drying trend in northwestern China, such as tree-ring based hydroclimate reconstruction (Liu et al 2019; Yang et al. 2019; Xu et al. 2014) and lake records (Lan et al. 2019)

References:

1. Shi, Y. et al. Recent and future climate change in northwest China. *Climatic Change* **80**, 379–393 (2007).
2. Yang, B., Wang, J. & Liu, J. A 1556 year-long early summer moisture reconstruction for the Hexi Corridor, Northwestern China. *Sci. China Earth Sci.* **62**, 953–963 (2019).
3. Liu, Y. et al. Anthropogenic aerosols cause recent pronounced weakening of Asian Summer Monsoon relative to last four centuries. *Geophys. Res. Lett.* **46**, 5469–5479 (2019).
4. Xu, G. et al. Drought history inferred from tree ring $\delta^{13}\text{C}$ and $\delta^{18}\text{O}$ in the central Tianshan Mountains of China and linkage with the North Atlantic Oscillation. *Theor. Appl. Climatol.* **116**, 385–401 (2014).
5. Lan, J. et al. Late Holocene hydroclimatic variations and possible forcing mechanisms over the eastern Central Asia. *Sci. China Earth Sci.* **62**, 1288–1301 (2019).

Lastly, it is absolutely essential that the authors share their isotopic measurement data, preferably on a modern platform like GitHub or at Zenodo since this is a routine exercise in transparency and reproducibility.

[Response]

The Nature Communications does not impose any specific platform requirements for data upload, such as GitHub or Zenodo. However, it mandates us to put the data in an accessible location for reviewers, and our current website effectively fulfills this requirement. Consequently, upon publication of our paper, we will ensure that the data are either uploaded on standard data repositories or made available through the NOAA paleodata website.

Reviewer #2 (Remarks to the Author):

The authors have address my minor concerns with the manuscript with the exception of one issue. They still include a "zero year" on the AD/BC timescale yet this year never actually existed. They quote a paper in support of this approach (Buentgen and Oppenheimer, 2020) but it is controversial. Many other established long chronologies are reported without this error. To avoid ambiguity, they should correct this error or state exactly what has been done.

[Response]

Thank you for pointing out this important comment. According to your suggestion, we have eliminated the term "zero year" in our data, and changed all "zero year" into (-1/1) in all relevant figures.

I will let the editor decide upon this minor concern. Otherwise, I happily recommend publication by Nature Communications.

[Response]

We sincerely appreciate your valuable suggestions and recognition of our work, which serve as a significant source of motivation and support for us.

REVIEWER COMMENTS

Reviewer #1 (Remarks to the Author):

Thank you for addressing my concerns, and the manuscript is much improved. Visual inspection finds the recent drought does not exceed the 95% confidence range of natural climate variability, indicating that it is not unprecedented in terms of statistics (Fig.3). The use of this word unprecedented in title and relevant text is overstated.

[Response]

Thank you for this valuable and important comment. In response, we have substituted the term 'unprecedented' with 'outstanding' in the title, as well as revised relevant sentences within the main text.

Figure R1 Visualization of the relationship between average tree-ring $\delta^{18}O$ values from this study with altitude is important, and it would be helpful if it is added in the supplement.

[Response]

Thank you for your suggestion. We have incorporated this figure into the supplement along with a corresponding description.

Reviewer #1 (Remarks to the Author): I am pleased the authors have answered my concerns. The manuscript is well-structured now, and can be accepted.

[Response]

Thank you for the suggestions which significantly enhanced the quality of our manuscript.